# InterCellar enables interactive analysis and exploration of cell–cell communication in single-cell transcriptomic data

Marta Interlandi [1,2 ✉], Kornelius Kerl [2] & Martin Dugas [1,3]

Deciphering cell–cell communication is a key step in understanding the physiology and pathology of multicellular systems. Recent advances in single-cell transcriptomics have contributed to unraveling the cellular composition of tissues and enabled the development of computational algorithms to predict cellular communication mediated by ligand–receptor interactions. Despite the existence of various tools capable of inferring cell–cell interactions from single-cell RNA sequencing data, the analysis and interpretation of the biological signals often require deep computational expertize. Here we present InterCellar, an interactive platform empowering lab-scientists to analyze and explore predicted cell–cell communication without requiring programming skills. InterCellar guides the biological interpretation through customized analysis steps, multiple visualization options, and the possibility to link biological pathways to ligand–receptor interactions. Alongside convenient data exploration features, InterCellar implements data-driven analyses including the possibility to compare cell–cell communication from multiple conditions. By analyzing COVID-19 and melanoma cell–cell interactions, we show that InterCellar resolves data-driven patterns of communication and highlights molecular signals through the integration of biological functions and pathways. We believe our user-friendly, interactive platform will help streamline the analysis of cell–cell communication and facilitate hypothesis generation in diverse biological systems.

[1] Institute of Medical Informatics, University of Münster, Münster, Germany. [2] Department of Pediatric Hematology and Oncology, University Children's Hospital Münster, Münster, Germany. [3] Institute of Medical Informatics, Heidelberg University Hospital, Heidelberg, Germany. ✉email: marta.interlandi@uni-muenster.de

Cell−cell communication plays a major role in multicellular organisms by driving cellular differentiation, contributing to tissue and organ homeostasis as well as coordinating both favorable and noxious immune responses in disease[1−4]. The advent of single-cell transcriptomics has offered the unprecedented opportunity to decipher the cellular heterogeneity of tissues and organs and, at the same time, investigate how different cell populations communicate at the molecular level. For example, a recent study examined the cellular composition of the human heart and revealed, through the analysis of cell−cell communication, that specific non-cardiomyocyte cells represent major communication hubs and might participate in maintaining heart contraction[5]. Moreover, due to the ongoing pandemic caused by the new coronavirus (SARS-CoV-2), multiple studies used single-cell transcriptomics to investigate which cell types might be more vulnerable to infection and how cellular communication might impact a positive or negative course of the disease[6−8]. In particular, these studies highlighted a central role of the immune system in the communication with epithelial cells of the lungs and illustrated how communication patterns might be distinguishing patients with mild symptoms from those who developed severe disease.

In the last few years, a number of methods have been developed to predict the occurrence of cell−cell communication by ligand−receptor interactions in single-cell RNA sequencing (scRNA-seq) data. In the seminal paper by Ramilowski et al.[9] the authors provided for the first time a manually curated collection of 2,422 ligand−receptor interactions that were used to predict cell−cell communication. Thereafter, extensive efforts have been made by the scientific community to improve such databases and to provide the necessary statistical framework to infer cellular communication between different cell types[10]. For example, more comprehensive ligand−receptor collections have been assembled through the inclusion of other molecular entities such as multi-subunit complexes and cofactors[11,12]. Several computational strategies rely on the co-expression of a given interaction pair (int-pair), evaluated on a cell cluster level. Specifically, for autocrine interactions, the expression levels of both genes (or gene complexes) comprising the int-pair are evaluated on the same cell cluster. On the contrary, for paracrine interactions, two different cell clusters are respectively tested for gene expression of the first and the second gene (or gene complex). Therefore, this type of inference of cell−cell communication relies on data pre-processing followed by clustering, with the aim of identifying relevant cell types that might take part in the communication[13]. Although many computational tools exist for the inference of cell−cell communication[10,13,14], the majority only provides limited downstream analysis functionalities, hindering the biological interpretation of the predicted interactions. Moreover, computational expertize is often required to visualize and produce interpretable results, thus representing a limiting factor for scientists with a deep biological background but basic programming skills.

Here we present InterCellar, an interactive platform implemented in R/Shiny that enables the downstream analysis and exploration of cell−cell communication based on scRNA-seq data. InterCellar's primary goal is to assist analysts in the biological interpretation of cellular interactions, by combining data-driven results with the possibility to customize the analysis. Requiring no programming skills, InterCellar is conceived as a final step in the analysis pipeline, fostering the collaboration between computational and non-computational scientists. Driven by a close collaboration with wet-lab scientists and physicians, we designed InterCellar as a user-friendly, dynamic application to streamline the analysis while removing the programming barrier existent for the majority of cell−cell communication inference tools.

The workflow of InterCellar builds upon pre-computed, inferred ligand−receptor interactions, which can be uploaded in the first step as a cell−cell interactions dataset (CCI data) (Fig. 1). The analyst has the option to provide either CCI data generated by InterCellar-supported tools (CellChat[12], CellPhoneDB[11], ICELLNET[15], or SingleCellSignalR[16]) or CCI data result of custom algorithms (e.g., Kumar et al.[17]) to predict cell−cell interactions (Supplementary Note 1). The second step offers multiple data exploration features and allows the user to focus the analysis on three biological domains, named universes: cell clusters, genes, and biological functions. In each universe (cluster-verse, gene-verse, and function-verse), different filtering and visualization options are provided, giving the opportunity to interactively subset the data and get qualitative and quantitative insights (Supplementary Note 2). For example, in the so-called function-verse, InterCellar provides multiple resources to perform functional annotation of enriched interactions. Lastly, InterCellar provides two types of data-driven analysis. The first enables a focused inspection of cell−cell communication through the analysis of the so-called int-pair modules, which are defined as groups of interactions with similar functional patterns and can be visualized and analyzed together with their significant functions. The second data-driven analysis implements functionalities to compare interactions across multiple datasets, highlighting patterns of communication that are uniquely found in each of the conditions considered. Importantly, InterCellar ensures reproducibility of the analysis and facilitates collection and finalization of results by providing multiple download options for tables and figures. In the following, we present InterCellar's main features and demonstrate its general applicability on two datasets, concerning melanoma and coronavirus disease 2019 (COVID-19).

## Results

**InterCellar allows data exploration through a user-friendly interface, customization options, and interactive visualizations.** Due to the high complexity of cell−cell interaction datasets, exploration of the data is crucial to get initial insights and comprehension of the communication patterns. For this reason, InterCellar's analysis workflow aims at exploring the data from different biological perspectives, while providing useful filtering options that allow the user to selectively refine the dataset.

In order to demonstrate InterCellar's exploration functionalities implemented in the three biological domains (universes), we considered a publicly available scRNA-seq dataset composed of metastatic melanoma samples from Tirosh et al.[18]. This dataset includes both malignant cells and cells from the tumor microenvironment (TME) of 19 patients. We retained the cell type labels assigned by the authors, namely melanoma-malignant cells, T cells, B cells, macrophages (Macro), endothelial cells (Endo), cancer-associated fibroblasts (CAF), and natural killer (NK) cells. To obtain inferred ligand−receptor interactions, we run CellChat[12], among InterCellar's supported input methods.

After uploading the predicted CCI data to InterCellar (Supplementary Fig. 1), the first biological domain of interest is represented by the cell clusters participating in the communication. These can be interactively explored in InterCellar's cluster-verse (Fig. 2a). The interface displays available filtering options, such as cluster selection or removal of interactions based on a minimum score or a maximum $p$-value. Filters adopted by the user automatically subset the original CCI data, updating InterCellar's outputs and getting propagated to further analyses. The cluster-verse offers three types of output: a network of cell clusters displaying the (total/weighted) number of interactions (overall or from a selected viewpoint cluster) (Fig. 2a), a barplot, and a table. To facilitate the inspection of large networks,

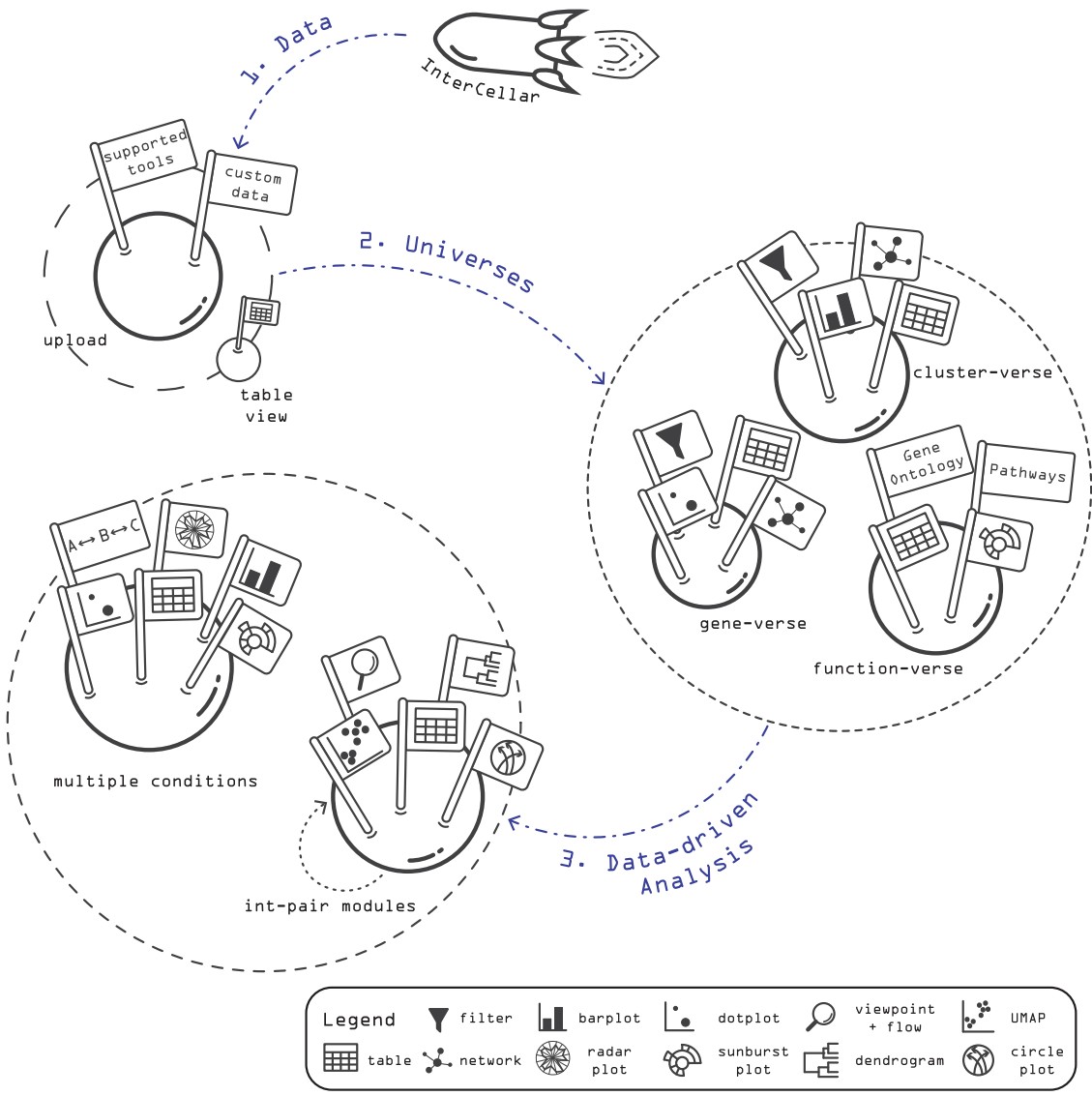

**Fig. 1 Representation of InterCellar's workflow.** The analysis workflow is composed of three main steps: (1) upload of user-provided data, containing predicted, pre-computed cell−cell interactions (CCI data); (2) exploration of three biological domains (so-called, universes), whereby the analysis focuses on cell clusters, genes, and biological functions; and (3) data-driven analysis through the definition and inspection of interaction-pair (int-pair) modules and the comparison of cell−cell communication from multiple conditions. For each step, filtering options (e.g., on cell clusters, genes, and functional databases) and multiple visualizations (e.g., networks, barplots, dotplots, circle plots, and sunburst plots) are provided to conduct a customized analysis. Moreover, all tables and figures can be saved in multiple formats, helping the interpretation of complex cell−cell interaction datasets and the finalization of publication-ready results.

InterCellar provides an interactive visualization that can be dynamically remodeled by the user with a simple drag-and-drop. Moreover, paracrine or autocrine interactions can be hidden from the visualization, and clicking on a cell cluster will highlight all the connected edges. For a better appreciation of InterCellar's interactive visualizations, we refer the reader to a video tutorial available at https://youtu.be/X5gUqzps4E4.

Proceeding in the data exploration, InterCellar's gene-verse focuses on the molecular components of the interactions, namely the proteins (and thus genes, in the case of scRNA-seq) that comprise ligand−receptor pairs. Once again, filtering options are available in this module and are specific for each input tool supported by InterCellar (Fig. 2b). In the present case, CellChat-specific filters regard the exclusion of pathways or annotation sources (see Supplementary Note 2). InterCellar's gene-verse offers the possibility to investigate precisely which cluster-pairs communicate through which ligand−receptor pairs. This is

achieved by manually selecting int-pairs of interest from a table (Fig. 2b, inset) which will trigger the generation of a dot plot (Fig. 2b) and a network. Here, for example, we selected all int-pairs that were annotated by CellChat as belonging to the TGFb signaling pathway. Customization options are available to the user (e.g., choice of color scheme, selection of cell clusters). Moreover, we enriched the information summarized in the gene-verse table, which lists only distinct int-pairs found in the CCI data, by providing links to Ensembl[19] and UniProt[20] databases to facilitate investigation of unfamiliar genes.

Lastly, we describe the functionalities implemented in Inter-Cellar's function-verse, which offers the opportunity to annotate interaction pairs with biological functions and pathways. In particular, InterCellar queries multiple integrated resources (Gene Ontology[21,22], KEGG[23], Reactome[24], Biocarta[25], PID:NCI-Nature[26], Panther[27], and PharmGKB[28]) that can be freely selected by the user in the interface (Fig. 2c). The annotated functional terms are then

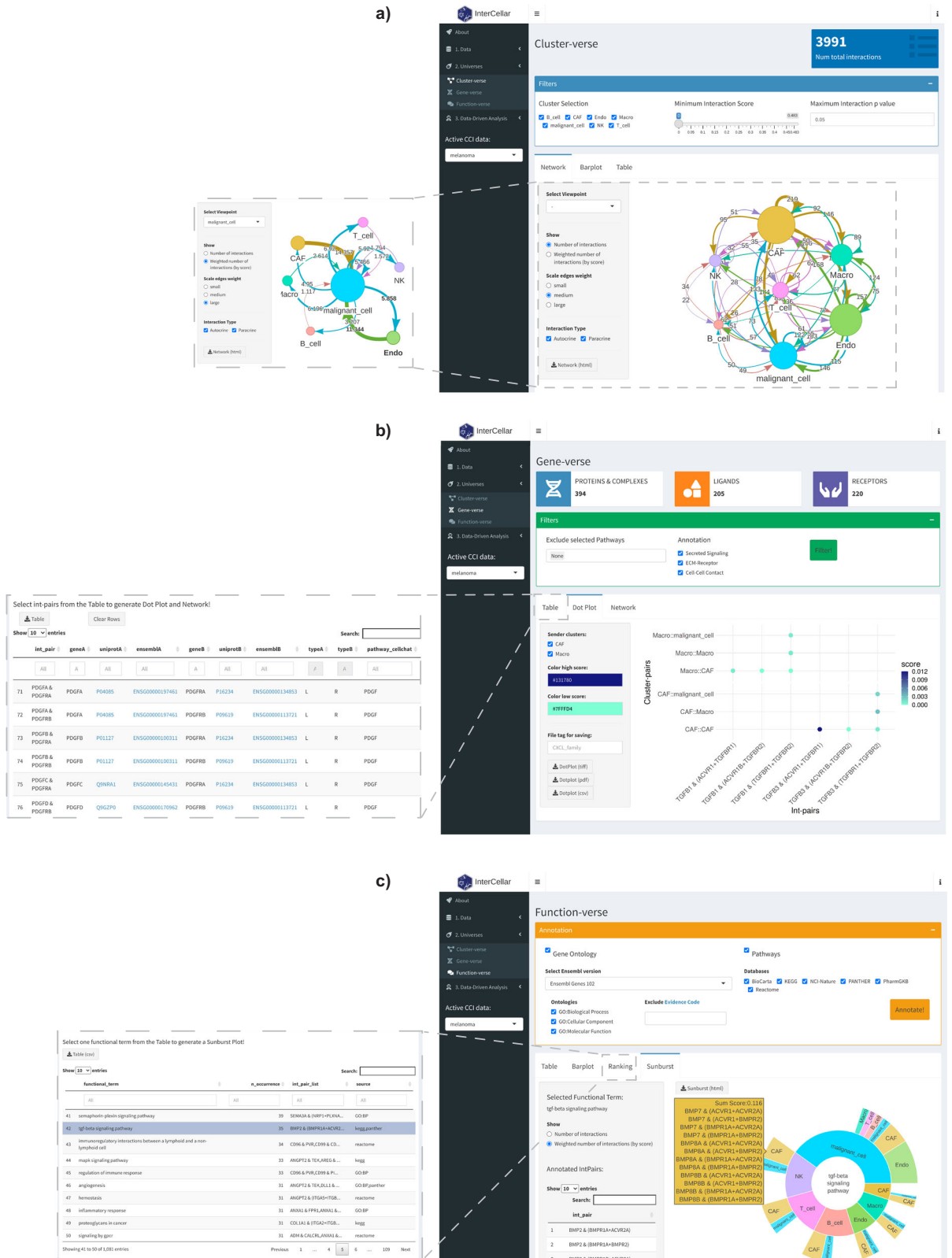

displayed in a table, summarized in a barplot, and ranked by the total number of int-pairs enriched. From this ranked table (Fig. 2c, inset), the user can examine functional terms of interest through a sunburst visualization, which combines all enriched int-pair/cluster-pair couplets (i.e., the occurrence of an int-pair in a certain cluster-pair). The sunburst visualization condenses multiple pieces of information in a single dynamic output and, to the best of our

knowledge, is a novel feature only provided by InterCellar among other CCI analysis tools. If we consider one int-pair (e.g., *BMP7* & (*ACVR1 + ACVR2A*)) enriched in our functional term of interest ("tgf-beta signaling pathway", annotated from KEGG and Panther), cell clusters expressing *BMP7* are represented in the inner ring (here, NK), while clusters expressing (*ACVR1 + ACVR2A*) are shown in the outer ring (here, CAF). The width of each section indicates the

**Fig. 2 Demonstration of InterCellar's features concerning data exploration. a** Screenshot of InterCellar's cluster-verse displaying a network of cell clusters. Filtering options are visible in the upper panel (cluster selection, minimum interaction score, maximum interaction *p*-value). In the lower panel, users can switch tabs among Network, Barplot, and Table. Network-specific customization options are available in the left panel as well as a download button to save the visualization. The network edges show the total number of interactions occurring between two cell clusters; all clusters are considered. The inset on the left-hand side shows a network generated by selecting malignant cells as viewpoint, with weighted numbers of interactions shown on the edges. **b** InterCellar's gene-verse is captured in a screenshot. CellChat-specific filtering options are shown in the upper panel, while the lower one displays a dot plot of selected int-pairs chosen by the user in the table (shown in inset). All int-pairs annotated by CellChat to the "TGFb signaling pathway" were selected. Customization and download options are available in the left panel. **c** Screenshot of the InterCellar function-verse. The upper panel provides multiple annotation resources that can be included/excluded by the user. Once the functional annotation is performed, the selection of one annotated functional term from the Ranking table (shown in inset) triggers the generation of a sunburst plot (here, "tgf-beta signaling pathway" annotated from KEGG and Panther). Hovering on an outer section of the sunburst plot will show a panel listing all int-pairs in the selected cluster-pair. A list of all int-pairs enriched is displayed on the left. In the same panel, the user can choose to generate the plot based on the total or weighted number of interactions. All visualizations are generated from CellChat-predicted CCI data of the melanoma dataset.

relative fraction of interactions (here, weighted by score) occurring in each cell type. Sections are arranged in descending order, for seamless identification of cluster importance within the functional term considered. Lastly, moving the mouse over each section provides detailed information on the interaction scores as well as a list of int-pairs enriched in the selected cluster-pair (Fig. 2c).

In conclusion, we have shown some salient features of InterCellar that aim at facilitating and streamlining the exploration of cell−cell interactions datasets. While some visualization outputs are conceptually similar to the ones generated by other analysis packages (see "Discussion"), InterCellar's interactive and intuitive interface, combined with dynamic features of visualization, offers a clear advantage to researchers without strong programming skills.

**InterCellar defines interaction-pair modules based on functional similarity.** The workflow of InterCellar combines data exploration with the possibility to obtain data-driven insights on cellular communication. Due to the intrinsically high redundancy that characterizes CCI data, where the same biological function can be orchestrated by a multitude of cognate ligand−receptor pairs, strategies to aggregate the data into intelligible information become crucial to an effective interpretation. InterCellar provides, to the best of our knowledge, a novel type of analysis based on int-pair modules, defined as groups of functionally similar interactions. Based on the functional annotation performed in the function-verse, InterCellar implements algorithms for dimensionality reduction and clustering that guide the user into a deeper analysis of the communication patterns, made biologically meaningful by the annotated functional terms. In order to demonstrate this step, we consider once again the melanoma dataset used thus far.

Since cancer cells have been described as actively recruiting and reprogramming normal cells of the tumor ecosystem[29], we used InterCellar to perform an in-depth analysis of the interactions that characterize malignant cells in their communication with the TME. Thus, we chose the malignant cell cluster as viewpoint in the analysis and focused on directed-outgoing interactions as those int-pairs where the ligand is expressed (and sent) by malignant cells to receptors expressed on all other cell types. After reducing the CCI data to the subset of interest, InterCellar uses the functional annotation to define int-pair modules: each module is composed of a subset of int-pairs that share common functional patterns. These modules can be visualized in a two-dimensional plot as a UMAP[30], which represents int-pairs clustered by functional similarity (Fig. 3a and Supplementary Fig. 2). Hence, a total number of nine int-pair modules was defined for the selected interactions and, for each module, we could associate functional terms found to be statistically significant (using a one-sided permutation test, see "Methods"). Moreover, InterCellar displays a circle plot to precisely identify

which cluster-pairs and genes comprise each module (Fig. 3b, c). Interestingly, by examining the circle plots, the relevance of many int-pairs could be validated by further literature research. For example, we investigated the underlying interactions for modules #6 and #8. Int-pair module #6, characterized by the functional terms "extracellular matrix (ECM) organization" and "integrin signaling pathways", is composed of collagen- and laminin-encoding genes expressed by malignant cells, which interact with integrin-complexes expressed by all other cell clusters, as well as malignant cells themselves. Notably, the importance of collagen-integrin interactions in promoting cell invasiveness has already been described in the literature by Zhou et al.[31]. Moreover, mechanisms of metastatic invasiveness in melanoma have been associated with the overexpression of annexin A1 in malignant cells[32]. Interactions involving this gene can be seen in the circle plot of int-pair module #8: *ANXA1*, sent by malignant cells, interacts with formyl peptide receptor (FPR) exclusively expressed by macrophages, suggesting a deleterious cross-talk between these two cell types.

Overall, we have shown that InterCellar implements state-of-the-art algorithms to group int-pairs into functional modules, reducing data complexity while preserving detailed biological information.

**InterCellar highlights data-driven patterns of cellular communication in the comparison of multiple conditions.** In addition to the parallel analysis of CCI data from multiple different conditions, InterCellar can automatically retrieve condition-specific communication patterns. The multiple conditions section of InterCellar, part of the data-driven analysis, serves this purpose. To illustrate functionalities and results, we consider a publicly available COVID-19 dataset from Chua et al.[6] containing nasopharyngeal and bronchial samples from 19 patients and from five healthy controls. In particular, we adopted the clinical classification of patients provided by the study, consisting of 8 moderate cases and 11 critical cases. Moreover, we retained the original cell type labeling performed by the authors and, as previously done by Chua et al.[6], ran CellPhoneDB (v2)[11] to obtain predicted cell−cell interactions as input to InterCellar. Thus, three separate input datasets were generated, corresponding to control, moderate and critical cases. Cell types can be grouped into epithelial cells and immune cells (see legend of Fig. 4).

As the first step, we considered the total number of interactions per cell type while comparing critical to moderate cases (Fig. 4a). We observed rather small differences in numbers of interactions for cell clusters belonging to epithelial cells (with the exception of ionocytes), while immune cell types showed a higher variability. In particular, we could confirm the findings of Lin et al.[7], who described a higher number of interactions for macrophages (MoMa and nrMa) as well as a lower number of interactions for

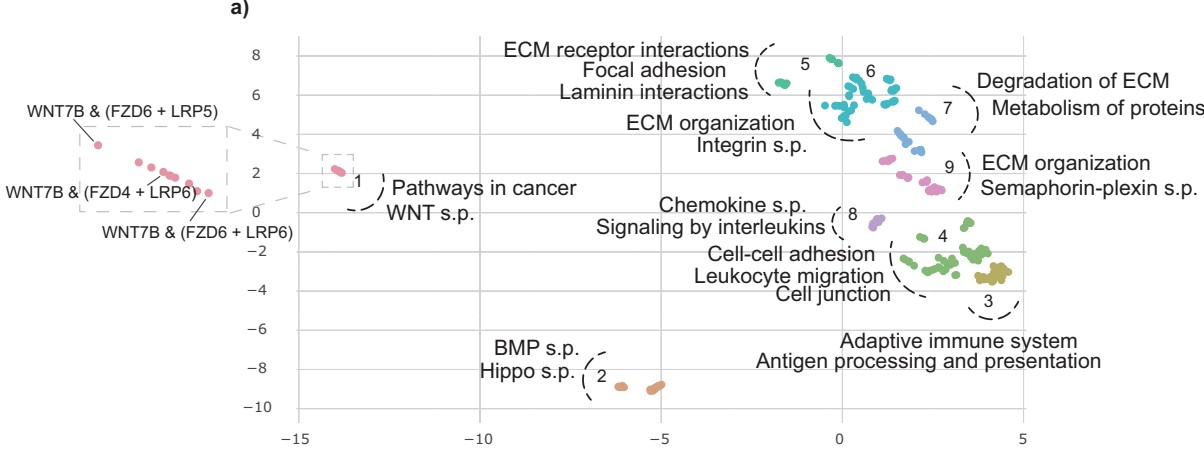

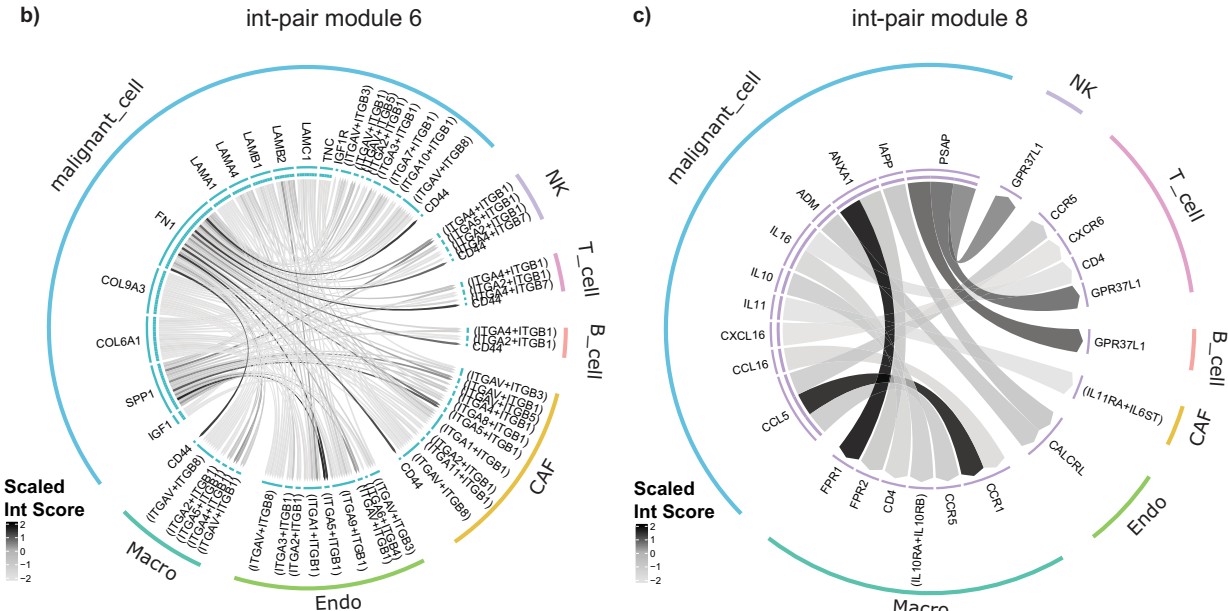

**Fig. 3 InterCellar drives the definition of interaction-pair modules uncovering distinct functional profiles of communication. a** UMAP of int-pairs enriched in melanoma malignant cells, when subsetting to directed-outgoing interactions. Each dot represents one int-pair, as shown, for example, in the inset corresponding to int-pair module #1. For each module, we manually selected functional terms from a list of significant terms (one-sided permutation test, $p < 0.05$) computed and displayed by InterCellar in a table. The selected terms serve as an example for identifying the biological functions of each module. **b**, **c** Circle plots for int-pair modules #6 and #8, respectively. Cell types are shown in the outer circle, while int-pairs are displayed in the inner one. Directional links originate from ligands and are pointing towards receptors; they are colored by a scaled (−2,2) interaction score to highlight relative differences in expression. s.p. - signaling pathway, ECM - extracellular matrix.

T cells (CTL and Treg), when comparing critical to moderate cases. Moreover, in the same comparison, comprehensive consideration of all cell types highlights a striking gain of interactions for mast cells (MC) as well as proliferating natural killer T cells (NKT-p). This finding holds true when examining the number of interactions between a certain cell type of interest and all other cell types (Fig. 4b and Supplementary Fig. 3a, b). Specifically, both immune cells (e.g., MoMa, nrMa, CTL, and Treg) and epithelial cells (e.g., ciliated, ciliated-diff, secretory and secretory-diff) show a consistent pattern of communication in critical cases, characterized by a higher number of interactions occurring between all cell types and MC or NKT-p cells. Interestingly, MCs have recently been hypothesized to have a major role in driving hyperinflammation in severe cases of COVID-19, due to their dysfunctional phenotype related to mast cell activation syndrome[33,34].

In conclusion, we have shown that InterCellar provides an unbiased and systematic approach to quantitatively compare cell−cell communication from multiple conditions.

**InterCellar unravels the composition of cellular communication by examining ligand−receptor pairs**. As previously described, another biological domain relevant for cell−cell communication is represented by the genes comprising the interactions. In particular, our objects of interest are int-pair/cluster-pair couplets, i.e., the occurrence of an int-pair in a certain cluster-pair. These couplets are visually represented by InterCellar in dot plots, where the analyst interactively chooses int-pairs and cluster-pairs to examine. Thus, we proceeded in the analysis of the COVID-19 datasets, by plotting the interaction score (calculated by CellPhoneDB, see "Methods") of

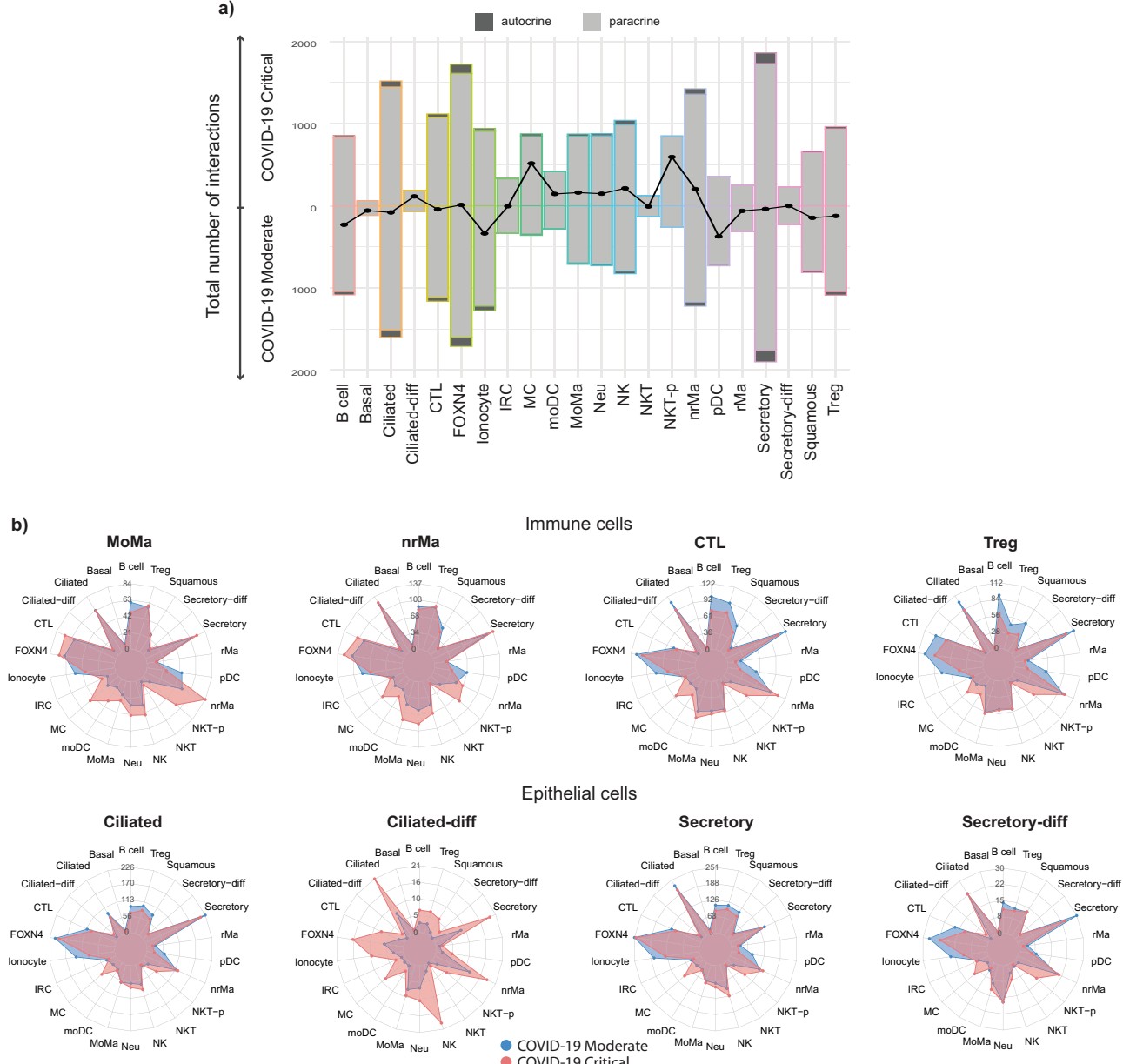

**Fig. 4 InterCellar generates comprehensive insights on the number of interactions per cell cluster. a** Differences in the total number of interactions per cell type are shown in back-to-back bar plots generated by InterCellar's multiple conditions section. All interaction flows are considered (directed-outgoing: L-R pairs, directed-incoming: R-L pairs, and undirected: L-L and R-R pairs). Dots on the black curve represent differences between the number of interactions in critical compared to moderate cases. Both autocrine and paracrine interactions are considered and plotted separately in each bar. Cell types are automatically color-coded by InterCellar for consistency throughout the analysis. See Supplementary Data 1 for source data. **b** Relative differences in the number of interactions are displayed in the InterCellar multiple conditions section using radar plots, comparing moderate to critical cases. All interaction flows are considered. Each plot displays the number of interactions between a specific cell type and all other cell types participating in the communication, ordered alphabetically, counterclockwise. Cell clusters belonging to the immune cell group are represented in the first row, while epithelial cell types are shown in the second row. Cell types can be grouped in (1) epithelial cells, namely basal, ciliated, ciliated-differentiating (ciliated-diff), FOXN4+, ionocyte, IFNG responsive cell (IRC), secretory, secretory-differentiating (secretory-diff) and squamous; and (2) immune cells, namely B cell, cytotoxic T cell (CTL), regulatory T cell (Treg), natural killer (NK), natural killer T cell (NKT), natural killer T cell-proliferating (NKT-p), neutrophil (Neu), plasmacytoid dendritic cell (pDC), monocyte-derived dendritic cell (moDC), mast cell (MC), resident macrophage (rMa), non-resident macrophage (nrMa) and monocyte-derived macrophage (MoMa).

int-pairs occurring in the communication between secretory cells, secretory-diff, and all other cell types. Here, we analyzed the three COVID-19 conditions in parallel, by using the dot plot functionality provided in the gene-verse. In particular, by looking at int-pairs composed of selected chemokine ligands mentioned in Chua et al.[6] (*CXCL1*, *CXCL3*, *CXCL6*, *CXCL16*, and *CXCL17*), we could observe a clear enrichment of ligand−receptor pairs in the communication

between secretory as well as secretory-diff cells and neutrophils (Fig. 5a). Specifically, these int-pairs were detected at varying levels of expression in the two disease conditions, while they were completely absent in control samples. This finding confirms the hypothesis of the authors concerning neutrophil recruitment induced by secretory cells[6]. However, no relevant difference could be recognized between moderate and critical cases.

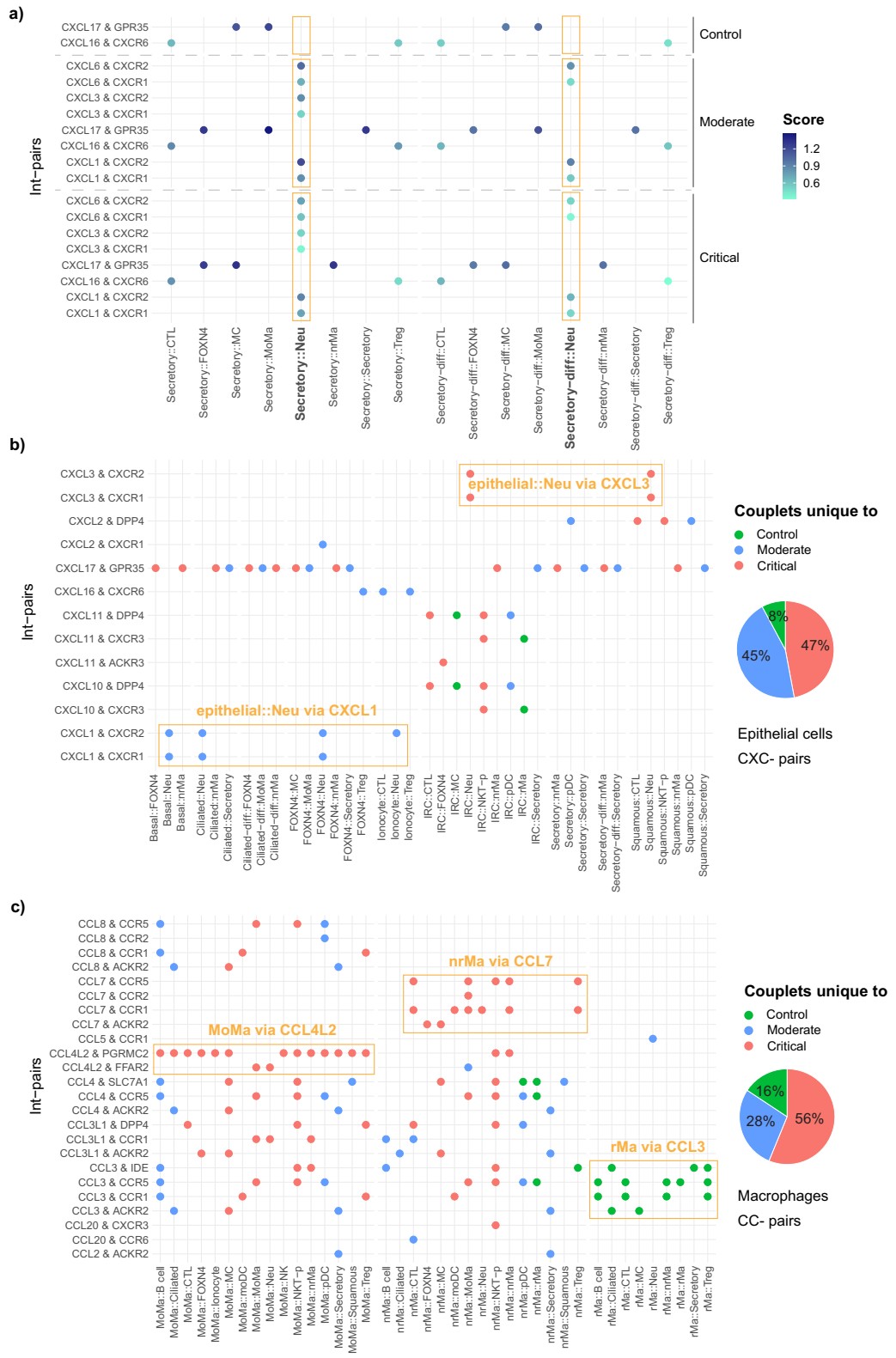

To further investigate the latter phenomenon, we made use of InterCellar's multiple conditions functionalities by considering only int-pair/cluster-pair couplets that are unique to a certain phenotype (i.e., control, moderate or critical). We chose two groups of chemokine ligands, namely the *CC-* and *CXC-* subfamilies, and selected all possible int-pairs found in each phenotype. *CXCL*-pairs showed a predominant enrichment of

interactions unique to critical cases, in both epithelial (47%) and immune cells (57%) (Fig. 5b and Supplementary Fig. 4a). Interestingly, due to this unbiased view of all int-pairs and cell clusters, we could notice, in moderate cases, neutrophil recruitment carried out by epithelial cells such as basal, ciliated, and *FOXN4*+ cells (through the pairs *CXCL1 & CXCR1* and *CXCL1 & CXCR2*). On the contrary, critical cases showed enrichment of

**Fig. 5 InterCellar allows in-depth analysis of interaction pairs and their enriched cell clusters. a** Selected chemokine pairs enriched in secretory and secretory-diff cells are shown in a dot plot, split by control, moderate and critical cases. Cluster-pairs are represented on the *x*-axis (e.g., cluster1::cluster2), while selected int-pairs are shown on the *y*-axis (e.g., geneA & geneB). Only statistically significant interactions are considered (one-sided permutation test by CellPhoneDB, *p* < 0.05). The Score represents the mean interaction score calculated by CellPhoneDB as the average of the mean expression of the interacting genes, evaluated in the respective clusters. Dot plots are generated in InterCellar's gene-verse. **b** Int-pairs belonging to the *CXC*-chemokine subfamily are evaluated in epithelial cell clusters. The dot plot represents only unique occurrences of int-pair/cluster-pair couplets, for each phenotype (control, moderate, critical). The overall contribution of each phenotype is summarized in a pie chart. Both dot plot and pie chart are generated in InterCellar's multiple conditions section. **c** Int-pairs belonging to the *CC*-chemokine subfamily are evaluated in macrophage cell clusters. As in **b**, only unique couplets are represented and overall contributions are summarized in a pie chart.

*CXCL3 & CXCR1* and *CXCL3 & CXCR2* in the communication from IRC or squamous cells to neutrophils (Fig. 5b). Regarding immune cells, critical cases were characterized by an outgoing cellular communication from moDC, MoMa, and NKT-p cells towards multiple other immune cell types. Moderate cases showed a unique pattern of communication promoted by NK cells and directed towards immune, as well as epithelial (secretory and ionocyte) cells (Supplementary Fig. 4a). For *CCL*-pairs, we focused on selected immune cell types, separated into three groups: macrophages (MoMa, nrMa, and rMa), NK cells (NK, NKT, and NKT-p), and T cells (Treg and CTL) (respectively Fig. 5c and Supplementary Fig. 4b, c). While for macrophages the proportion of unique interactions favors the critical phenotype (56%), an inverse tendency could be noticed for NK (41% critical) and T cells (20% critical). In particular, among macrophages, MoMa and nrMa displayed many chemokine interactions that were unique to critical cases. These interactions involved recipient cell types such as MC, moDC, neutrophils, and NKT-p. Specifically, MoMa communicates with these other cell types via *CCL4L2* while nrMa via *CCL7*. At the same time, interactions of rMa via *CCL3* in controls are absent in COVID-19 cases (Fig. 5c). On the contrary, NK and NKT as well as Treg and CTL showed an enrichment of interactions unique to moderate cases, directed towards secretory cells (among others) (Supplementary Fig. 4b, c). Altogether, these results are in line with two findings by the authors: on the one hand, critical cases are characterized by a highly inflammatory profile for MoMa and nrMa; on the other hand, a well-balanced immune response distinguishes moderate cases, in which the communication between immune cells and epithelial cells underlie an effective response to the viral infection[6].

In summary, InterCellar provides valuable features to perform an in-depth investigation of the communication signals carried out by selected interaction pairs and their enriched clusters. When comparing multiple conditions, InterCellar automatically highlights the occurrence of int-pair/cluster-pair couplets that are condition-specific, thus facilitating the detection of communication patterns that distinguish each phenotype.

**InterCellar uncovers condition-specific interactions and their related biological pathways**. As a final step in our analysis, we present the functionalities implemented in InterCellar's multiple conditions section that are based on the aforementioned functional annotation. Here, specifically, InterCellar determines which int-pairs are occurring uniquely in each condition (independently from the cluster-pairs) and, via a permutation test, identifies significant functional terms annotated to the unique sets of int-pairs. Thus, the analyst can promptly investigate condition-specific interactions and their related biological pathways that might be dysregulated between the conditions of interest. Once again, we show InterCellar's results based on the comparison between COVID-19 critical and moderate cases. Firstly, the functional annotation was performed for each dataset, using all functional databases provided by InterCellar. Then, for each

condition, significant functional terms of interest could be visualized in a sunburst plot. In contrast to the sunburst plots available in the *function-verse*, the present visualization only considers int-pairs that uniquely characterize the chosen condition. Interestingly, the functional term "inflammation mediated by chemokine and cytokine signaling pathway" was found to be significant in critical COVID-19 cases, supporting the previous hypothesis derived from the analysis of chemokine and, once again, delivering signals of a cytokine storm[35] (Fig. 6a). On the contrary, among significant functional terms describing moderate COVID-19 cases, we found "T cell costimulation" (Fig. 6b). As previously mentioned, sunburst plots offer the advantage of immediately delivering insights regarding cluster importance within the functional term considered. For moderate cases, CTL are by far the main partners in interactions involved in "T cell costimulation", which, in this specific case, is regulated by only one condition-specific int-pair, namely *CD160 & TNFRSF14*. As for critical cases, we noticed that nrMa contributes to almost half of the total interaction score associated with the term. Furthermore, they communicate to other cell clusters through condition-specific int-pairs of the chemokine family, which were already highlighted by the dot plot of the previous analysis (Fig. 5c). Overall, these results suggest once again a dysfunctional immune response in COVID-19 critical cases, where a balanced lymphocyte-mediated regulation seems to be overthrown by excessive activation of macrophages.

Finally, we have shown that InterCellar provides a straightforward and effective strategy to reveal likely dysregulated biological pathways across conditions.

## Discussion

In the context of single-cell transcriptomics, the investigation of cellular cross-talk has been regarded as a valuable method for understanding biological mechanisms in health and disease[10]. Despite the existence of many algorithms for predicting the occurrence of cell–cell communication based on ligand–receptor interactions, further analysis and interpretation of the associated biological signals are non-trivial due to the complexity of results and the lack of a standardized approach to perform such analyses[13].

We developed InterCellar as an interactive analysis platform designed to guide scientists in the interpretation of complex cell–cell communication results obtained from scRNA-seq data. InterCellar's accessibility and ease of use, along with customization options and information-rich visualizations, can help streamline the analysis of cell–cell communication by promoting collaboration between wet-lab and computational scientists. By demonstrating InterCellar's functionalities on two different scRNA-seq datasets, we were able to obtain comprehensive insights on the modalities of cell–cell communication and the underlying ligand–receptor interactions, which could be further corroborated by literature review. Moreover, we have shown that InterCellar is a powerful tool for identifying and highlighting previously unknown molecular interactions inferred from single-cell transcriptomic

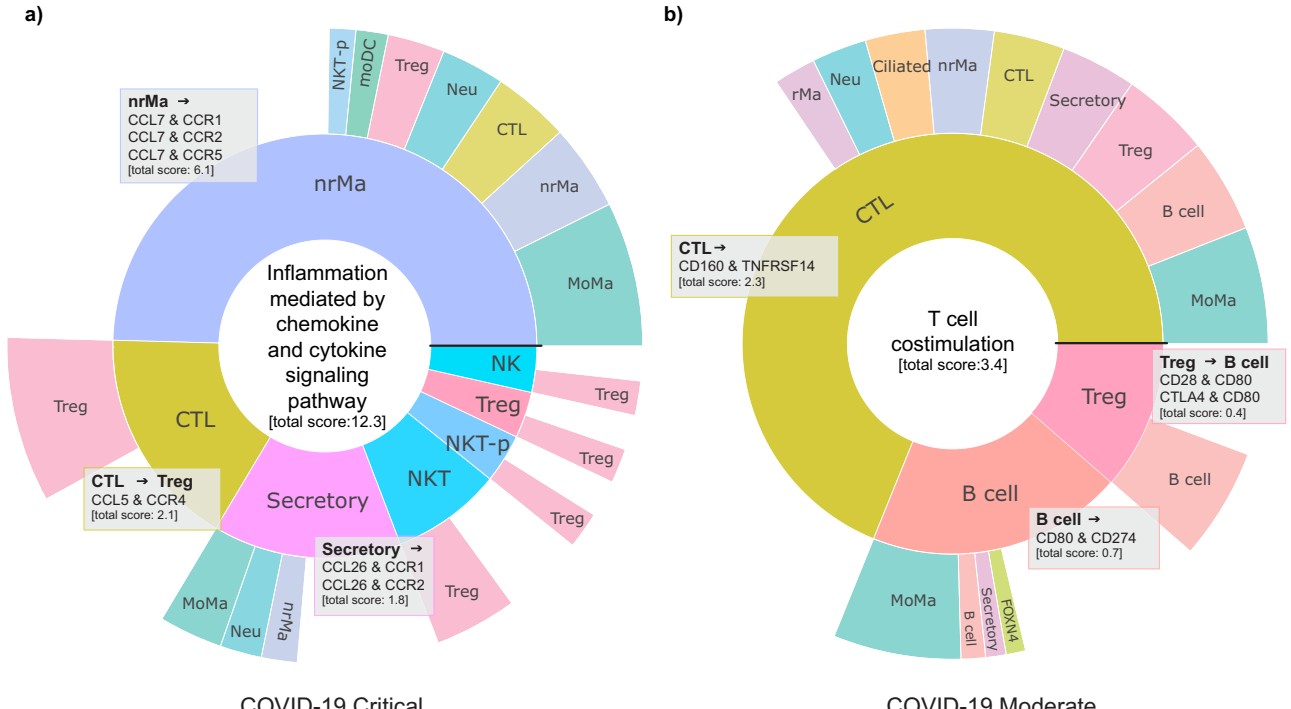

**Fig. 6 Investigation of significant functional pathways and their enriched condition-specific interactions.** Sunburst plots for two condition-specific functional terms found to be significant (one-sided permutation test, *p* < 0.05) in the comparison between **a** COVID-19 critical ("Inflammation mediated by chemokine and cytokine signaling pathway") and **b** moderate cases ("T cell costimulation"). Plots are generated in InterCellar's multiple conditions module, by selecting the chosen terms from a table. Only int-pairs unique to each condition and annotated to the selected functional term are considered in the generation of the plot. For a generic int-pair 'geneA & geneB', enriched in 'cluster1::cluster2', the first cell cluster (cluster1, expressing geneA) is represented in the inner circle, while the second (cluster2, expressing geneB) is shown in the outer circle. The width of each section indicates the relative fraction of interactions (weighted by score) enriched in that cell type. Sections are arranged in descending order, counterclockwise, starting from the horizontal black line. The total interaction scores per cell type are reported in brackets. Gray boxes show the list of condition-specific int-pairs enriched for the corresponding cluster-pairs and were manually added to the figure from the downloaded files provided by InterCellar.

data. For example, an in-depth analysis of cellular interactions in COVID-19 patients confirmed the primary role of immune response in discriminating between different courses of the disease[6]. In addition, the analysis suggested a consistent pattern of communication for patients with critical COVID-19, involving cell types such as mast cells. These results may provide exciting hypotheses to deepen the understanding of severe SARS-CoV-2 cases.

Implemented in R/Shiny, InterCellar is distributed as an open-source Bioconductor[36] package, ensuring software robustness, maintenance, and compatibility with multiple operating systems. With an existing installation of R and Bioconductor on the user's personal computer, installing the InterCellar package and launching the Shiny app can be accomplished by running two lines of code, which, together with further information, can be found in the online user guide (https://bioconductor.org/packages/release/bioc/vignettes/InterCellar/inst/doc/user_guide.html). Alternatively, InterCellar is available in a docker container and as a bioconda recipe (http://bioconda.github.io/recipes/bioconductor-intercellar/README.html). Notably, the local installation of InterCellar avoids issues related to data privacy or sharing of unpublished patient data, which is a typical problem for web-based platforms running on external servers. Moreover, each step of InterCellar's workflow is generally completed in a very short time (from seconds to few minutes), ensuring a truly interactive and performant analysis.

InterCellar directly builds upon the results of other existing tools developed to infer cell–cell communication, which can be freely chosen by the user. Alongside the automatic import from supported tools, InterCellar accepts custom data with specific

format and necessary information (see "Methods"). Since prediction methods often rely on different reference databases and build their results on diverse statistical and mathematical assumptions, evaluating advantages and disadvantages of each method is critical for choosing the method that best fits the data of the user[10,14]. Lastly, we want to stress the importance of further experimental validation of the predicted interactions, which could be achieved, for example, by single-molecule fluorescence in situ hybridization or measurement of co-occurrence through flow cytometry[10].

To benchmark InterCellar's features, we identified 18 published tools from two recent review papers[10,13], as well as from manual search (as of April 2021, Supplementary Table 1). We collected methods allowing the analysis of cell–cell communication and excluded 13 methods that required programming skills (e.g., extensive coding in R/Python to perform the analysis) from a systematic comparison, thus focusing on the remaining web-based or standalone systems (Supplementary Fig. 5). We found that one tool, CCCExplorer[37], had comparable functionalities to InterCellar, providing a local, interactive analysis that focuses on enriched biological pathways. However, this software was developed for bulk RNA sequencing, therefore lacking a straightforward application in scRNA-seq. Four platforms, namely pyMINEr[38], talklr[39], CellChat Explorer[12], and Cellinker[40] are limited regarding interactive analysis, customization, and download capabilities; therefore, these tools could be viewed primarily as explorative tools that provide static results. Moreover, CellChat Explorer, talklr, and Cellinker run remotely on a server, thus requiring data sharing and possibly leading to privacy issues.

The authors of CellChat recently provided a Shiny app with the same set of functionalities as CellChat Explorer and the advantage of local installation through a docker container. Lastly, in the interest of those readers with basic programming skills, we performed a detailed comparison between InterCellar and the Cell-Chat R package and outlined differences and similarities in Supplementary Note 3 (see also Supplementary Figs. 6, 7).

In the future, we plan to further extend the input options to other published tools. This might, on the one hand, facilitate internal comparison and validation of biological conclusions drawn from interactions predicted by different methods and, on the other hand, give more flexibility to the end-user.

In conclusion, InterCellar empowers lab scientists to interactively analyze cell−cell interactions without programming skills needed. Moreover, it implements data-driven approaches to aggregate interactions into functionally distinct modules and to automatically discern interactions and pathways that are specific to a certain condition. By providing a comprehensive workflow, along with several graphical and dynamic outputs, InterCellar facilitates the simplification and standardization of the downstream analysis of cell−cell communication. We believe Inter-Cellar will contribute to accelerating the generation of hypotheses and deepening the understanding of the cellular cross-talk in various pathological and physiological systems.

## Methods

**Implementation**. InterCellar is implemented as an R/Shiny application and structured as an R package, which is available on Bioconductor at https://bioconductor.org/packages/InterCellar/. We used the R package golem to ease the development of a robust Shiny application, as well as multiple R packages to build the user interface (shiny, shinydashboard, shinyFiles, shinycssloaders, shinyFeedback, shinyalert, htmltools, and htmlwidgets). Data handling is achieved through custom R functions that build upon public R/Bioconductor packages, such as readxl, utils, DT, plyr, dplyr, tidyr, and tibble. The functional annotation of InterCellar's function-verse is performed through two R packages that query functional databases, namely biomaRt[41] and graphite[42]. Lastly, graphical outputs are generated by custom R functions based upon R/Bioconductor packages such as ggplot2, plotly, circlize[43], fmsb, umap, visNetwork, igraph, dendextend[44], factoextra, colourpicker, scales, and grDevices.

**Input tools and preprocessing**. InterCellar's input data consists of a pre-computed dataset of predicted cell−cell interactions (CCI data). The application accepts either the output of supported tools (CellChat[12], CellPhoneDB[11], ICELLNET[15], or SingleCellSignalR[16]) or of custom analyses performed by the user (for example, as in Kumar et al.[17]). In the first case, InterCellar automatically parses the output generated by the supported tool, requiring no further data manipulation by the user. For a detailed description of the expected input data, depending on the supported tool, we refer to Supplementary Note 1 (see also Supplementary Tables 2−5). In the second case, the custom input data must contain relevant information structured as a table with the following columns: (i) interaction pairs, containing the names of two molecular components that participate in the interaction (e.g., compA_compB); (ii) communicating cell clusters, containing names or numbers of the cell populations (divided into two columns, e.g., clustA and clustB); (iii) molecular types, either ligand or receptor (in two columns, corresponding to compA and compB); and (iv) a numeric value representing a score for each interaction (e.g., average expression of compA_compB over clustA and clustB) (see Supplementary Fig. 1, inset).

For both input options, InterCellar will preprocess the data to generate a standardized dataset. These preprocessing steps involve: (1) mapping int-pairs to the associated gene symbols; (2) annotation of the molecular type of each int-pair by combining the provided information with a manually curated set of int-pairs (available at https://github.com/martaint/InterCellar-reproducibility); and (3) reordering int-pairs listed as receptor-ligand to ligand−receptor, to ensure consistency in the definition of the communication flow. To this date, InterCellar supports human genes; the analysis of cell−cell communication obtained from other species can be performed by converting gene names to human orthologs.

**Filtering and customization**. We implemented multiple filtering options to enable a flexible interactive analysis of cell−cell communication. In particular, we distinguish between two types of filters: those applied to the input data and filters applied to visualization options. In the first case, the user can (i) remove entire clusters from the analysis, for example in the event of unknown or poorly defined cell populations (in cluster-verse); (ii) refine int-pairs selecting either a minimum interaction score or a maximum p-value (in cluster-verse); (iii) select int-pairs by

criteria that are specific to the supported tool used to generate the input data (in gene-verse). These three filtering options are applied to the input dataset; therefore the following analyses will be performed on the data subset. For a detailed description of filtering options specific to the gene-verse, we refer to Supplementary Note 2. The second type of filters concerns graphical outputs and includes features to customize visualizations. These options are only applied to the selected plot and will not affect the underlying dataset. For example, a dot plot visualization available in the gene-verse displays int-pair/cluster-pair couplets based on the user's selection of int-pairs of interest. The user can further select a subset of clusters to consider, as well as change the color scheme of the dot plot.

**Functional annotation of interaction pairs**. InterCellar provides interactive annotation of int-pairs with the aim to link biological functions and pathways to cell−cell communication. Specifically, we use two Bioconductor packages, biomaRt[41] and graphite[42], to automatically query existing databases of functional terms: Gene Ontology[21,22], KEGG[23], Reactome[24], Biocarta[25], PID:NCI-Nature[26], Panther[27], and PharmGKB[28]. InterCellar's functional annotation (available in the function-verse) is required before proceeding to the data-driven analysis step. Importantly, the annotation of a certain functional term to an int-pair is fulfilled only when all components (e.g., ligand and receptor genes) of the int-pair are enriched by that functional term. This applies also in the case of multi sub-unit complexes (e.g., for CellPhoneDB and CellChat), where all components of a complex must be enriched by a functional term. Although the user can freely choose which functional sources to consider when performing the annotation, we suggest including as many databases as possible to maximize the number of int-pairs which will be annotated by at least one term. Only these annotated pairs are further considered in the definition of int-pair modules.

**Definition of interaction-pair modules**. Defining int-pair modules requires three key, user-driven decisions: (1) choice of the viewpoint cell cluster, representing the cell type of interest; (2) selection of a communication flow, among directed-outgoing (in which the viewpoint cluster sends ligands to other clusters), directed-incoming (in which the viewpoint cluster expresses receptors) and undirected (as in the case of receptor-receptor pairs); and (3) definition of the number of int-pair modules to consider. In particular, InterCellar subsets the result of the functional annotation (i.e., a binary matrix of int-pairs by functional terms) according to steps (1) and (2) (Supplementary Fig. 2). This filtered matrix is given as input to a dimensionality reduction algorithm called UMAP (Uniform Manifold Approximation and Projection[30]), which calculates a 2D-embedding of the interaction pairs considered (using cosine similarity as metric to compute distances between data points). Thus, the low-dimensional embedding reflects the similarity of int-pairs based on their functional profiles. Finally, using the UMAP coordinates, a hierarchical clustering defines modules of int-pairs that share similar functional profiles (with euclidean distance and ward.D2 clustering algorithm). The UMAP and an additional dendrogram result of hierarchical clustering can be visualized and downloaded. Furthermore, two plots provide guidance on choosing the optimal number of int-pair modules: (i) total within-cluster sum of squares (WSS) (i.e., elbow method) and (ii) average silhouette width. Both metrics are computed and visualized using functions provided in the R package factoextra. The first metric indicates the compactness of clusters, by looking at intra-cluster variation. Specifically, one seeks to minimize the total WSS, represented as a function of the number of clusters. The elbow method, in particular, defines the optimal number of clusters by looking at a bend (i.e., the elbow) in the plot. We automated this process by using the R package akmedoids, which provides a function to find the maximum curvature. The average silhouette width evaluates how well each data point lies within its cluster and, on the contrary, should be maximized. By default, InterCellar uses the optimal number of modules computed by the elbow method to generate UMAPs and dendrograms; this value can however be freely changed by the user. When the total number of unique int-pairs considered (by choosing viewpoint and flow) is less or equal to 10, only one module is defined to prevent noisy results due to insufficient data points.

**Statistics and reproducibility**. To facilitate the biological interpretation of the defined int-pair modules, InterCellar calculates an empirical p-value for each annotated functional term indicating statistical significance of a term to a certain int-pair module. To this end, we implemented a one-sided permutation test which considers as test statistic the proportion of int-pairs annotated to each term, per module (called module ratio). By randomly shuffling the module assignment of each int-pair (1000 times, without replacement) and calculating the test statistic, we generate the distribution under the null hypothesis. Finally, we compute for each functional term the proportion of the module ratios which are higher or equal to the actual module ratio, thus obtaining an empirical p-value for the specificity of a functional term to a certain int-pair module. Functional terms are then ranked by p-values, and the threshold for significance can be chosen by the user (0.05 by default). As for the comparison of multiple conditions, InterCellar computes significant functional terms associated with each condition using the same statistical method delineated above, replacing the module-assignment of each int-pair with a condition-assignment.

The analysis presented in this study can be reproduced following walk-through tutorials available at https://github.com/martaint/InterCellar-reproducibility. Moreover, this repository contains the code used to generate InterCellar input data for the two datasets analyzed (COVID-19 and melanoma).

**Melanoma dataset**. For the melanoma dataset, a preprocessed gene expression matrix was downloaded from NCBI Gene Expression Omnibus (GEO) (GSE72056), containing transcript-per-million gene expression values for a total of 4,645 cells, comprising both melanoma malignant cells and cells from the tumor microenvironment. We removed ~11% of these cells, due to a missing or unknown cell type label, retaining 4,097 final cells. We run CellChat[12] (version 1.1.2) to obtain predicted cell−cell interactions.

**COVID-19 dataset**. This dataset comprises single-cell RNA sequencing data of 24 total patients, divided into five healthy controls, eight COVID-19 moderate cases, and 11 COVID-19 critical cases. A total number of 32 samples were derived from nasopharyngeal-protected specimen brush and bronchial lavage. Classification in moderate and critical cases was performed by the original authors following WHO guidelines. The authors stated that signed informed consent was obtained from all patients before inclusion in the study. Furthermore, their study was approved by the respective institutional ethics committee of either the Charité-Universitätsmedizin Berlin (EA2/066/20) or the University Hospital Leipzig (123/20-ek) and conducted in accordance with the Declaration of Helsinki[6]. Pre-processed, normalized data were retrieved from Chua et al.[6], as well as cell type assignment. We removed two cell clusters whose label assignment was poorly defined, namely "unknown epithelial" and "outlier epithelial", corresponding to ~1.5% of the total number of cells. Moreover, moderate and critical datasets were randomly subsampled to 10,000 cells each (without losing any cell label), while for the control dataset we retained all cells, corresponding to a total of 2,966 cells. Finally, for each of the three datasets, we run CellPhoneDB[11] v2 statistical analysis using default parameters.

**Reporting summary**. Further information on research design is available in the Nature Research Reporting Summary linked to this article.

## Data availability
All data analyzed in this study are publicly available. In particular, for the COVID-19 datasets[45], we retrieved preprocessed data for the three conditions (control, moderate and critical) from FigShare at https://doi.org/10.6084/m9.figshare.12436517.v2 (data object named covid_nbt_main.rds). For the melanoma dataset[46], preprocessed data were downloaded from GEO with Accession Number GSE72056. InterCellar input datasets used in this study are available at https://github.com/martaint/InterCellar-reproducibility. Source data underlying Fig. 4a are presented in Supplementary Data 1. All other relevant data supporting the key findings of this study are available within the article and its Supplementary Information files or from the corresponding authors upon reasonable request.

## Code availability
The InterCellar package (version 2.0.0) is available on Bioconductor v3.14 (requiring R4.1 or higher). Alternatively, the package can be installed following the instructions on the GitHub repository (https://github.com/martaint/InterCellar), which contains source code and user guidelines. The source code is also citable, as deposited in Zenodo[47].

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

## Acknowledgements

We would like to acknowledge and thank members of the K.K. lab, as well as members of IMI, for their insightful comments and suggestions. A special thanks to our colleagues Carolin Walter and Thomas K. Albert for their critical reading of the manuscript; to Flavia Watusi de Faria, Viktoria Melcher, Monika Graf, Natalia Moreno, and Teresa Gerling for testing InterCellar on different datasets and giving feedback for improvements. M.D. is supported by funds from European Union grant MDS-RIGHT [634789]. K.K. is supported by funds from the Deutsche Krebshilfe (70113653), the Deutsche Forschungsgemeinschaft (KE 2004/4-1) and the Wilhelm Sander Stiftung (2019.158.1).

## Author contributions

M.I., K.K., and M.D. conceived the project. M.D. supervised the bioinformatics part of the research while K.K. supervised the biological aspects. M.I. developed and implemented the software, analyzed the data, and prepared the figures. M.I., K.K., and M.D. wrote the paper. All authors read and approved the final manuscript.

## Funding

## Competing interests

The authors declare no competing interests.
