## [Transparent Peer Review File · Communications Biology]

Reviewers' comments:

Reviewer #1 (Remarks to the Author):

The authors present an interactive R/Shiny package, InterCellar, for downstream analysis of cell-cell communication inference (CCI) derived from scRNA-seq data. The goal of InterCellar is to analyze the data over three biological domains, referred to as "verses": the (cell) cluster-verse, gene-verse, and the function-verse, which seeks to infer biological function via functional annotation of inferred interactions. In cluster-verse, InterCellar visualizes results with respect to the total number of interactions per cell type; in the gene-verse, InterCellar investigates which cluster-pairs communicate through which genes; and in the function-verse, it annotates cell-cell communication with biological functions implied by enriched pathways. The authors demonstrate its main features by applying it to two datasets of COVID-19 and melanoma from the literature. Using InterCellar, the authors present results that are consistent with previous findings; they also present some novel findings on cell communications in the two datasets. By comparison with similar packages, the authors claim that InterCellar has the following advantages: It provides a local and interactive analysis that focuses on enriched biological pathways without programming skills, and the workflow can be run over a relatively short time frame.

The manuscript is well written, particularly the Introduction. The figures are nicely done, and the motivation to streamline CCI analysis from single-cell RNA-seq data is worthwhile. Moreover, the functional analysis presented is a very interesting way to interpret CCI output. However, it is not fully clear what novel features InterCellar presents that are not available in other packages, such as CellChat. Moreover, the authors attribute many novel results not presented in previous studies to InterCellar's features, but one could arguably attribute the results to CellPhoneDB, which was applied to each dataset. Overall, there is a lack of explanation of the underlying methods used during the downstream analysis and the arguments for why InterCellar should be adopted over other methods, particularly CellChat, could be fleshed out.

Therefore, I recommend acceptance only after major revisions.

Below are more comments:

- It is great that the authors allow for flexibility of various CCI outputs as input for InterCellar. However, it is not entirely clear what type of data is needed. Is it a matrix, or an R dataframe? Clarifying this would strengthen the benefits of InterCellar more.
- There is a lot of overlap between InterCellar and CellChat, in terms of the visualization capabilities. The way that the results are presented in the manuscript imply that InterCellar is a way to extract more meaning from results from packages like CellPhoneDB, as is done for CellChat. I think it would be worth fleshing out a comparison between the two methods to better highlight how InterCellar can be used for downstream analysis.
- The authors rightfully stress the importance of reducing the "programming barrier" required to make better use of CCI packages. However, it is not clear how InterCellar is easier than other methods, as all cell-cell communication methods require some degree of programming expertise. If researchers are able to run cell-cell communication analysis and analyze single-cell RNA-seq data, then it is not unreasonable to expect that they know how to visualize the communication output (with help from tutorials).
- The filtering/visualisation steps of InterCellar are very nice but there's not much mention of them in the main results text. It would be interesting to see how the filtering used by InterCellar affects downstream analysis.
- It is unclear whether the figures shown in the manuscript are produced near-automatically from the R/Shiny workflow of InterCellar, or if these are polished figures based on raw output from InterCellar. If the former is true, this should be highlighted more to strengthen the novelty of InterCellar. If the latter is true, then this is somewhat contradictory to the aims of InterCellar to lower the programming

expertise barrier.

- Counting only number of interactions/interaction pairs assumes that each pair has equal effect on communication. Have the authors considered weighting the interactions by interaction score, i.e. instead of number of interactions, you count total interaction score, and instead of Figure 4 counting the fraction of numbers, you could the fraction of total weight?
- The function-verse is perhaps the most novel feature of InterCellar and is a very nice way to interpret CCI output. I think the descriptions underlying the function-verse-specific methods could be expanded more to highlight its novelty and usefulness.
- The authors use the function-verse outputs to classify interactions based on functional similarity. The term "functional similarity" is also used in CellChat for downstream analysis, albeit using a different methodology. Again, it would be useful to highlight the differences between InterCellar and CellChat for functional analysis.
- As the authors run CellPhoneDB for each of the datasets considered, it is hard to attribute any of the new biological results to InterCellar specifically, as there is no benchmark of comparison. Moreover, none of the original papers had run CellPhoneDB, so it is not entirely clear whether InterCellar would have found new results that would not have been found if users had just applied CellPhoneDB on its own. Is it possible to analyse a dataset where the CCI has already been run, where the authors do not need to run CellPhoneDB?
- The authors state that InterCellar accepts input from SingleCellSignalR and custom CCI input. However, the results presented only consider CellPhoneDB-derived output. How do the results change if SingleCellSignalR or other CCI output is used instead?
- CellPhoneDB handles multiple ligand/receptor sub-units but it looks like InterCellar does not. How does InterCellar reconcile this?
- I see that on BioConductor, InterCellar requires R 4.1, but in the manuscript, the authors claim that there are installation instructions for R 4.0.X at the GitHub repository. However, I could not find these instructions.

Reviewer #2 (Remarks to the Author):

Existing tools for inferring cell-cell communication often require computational expertise to interpret biological signals, which limits scientists without programming skills to easily analyze, explore and interpret predicted intercellular communication. In this work, Interlandi, Kerl and Dugas present InterCellar, an interactive platform intended to fulfill this gap and facilitate the biological interpretation through customizable analyses and visualization options to easily identify important ligand-receptor interactions and meaningful associated biological pathways. The authors also provided COVID-19 and cancer examples to demonstrate what InterCellar offers for achieving this purpose. This work claims to empower lab-scientist without programming skills to analyze and explore results from inferred cell-cell communication. Meeting this intention is an important contribution to the cell-cell communication field, which would clearly help massify these analyses and have a better understanding of associated biological processes, this this work will be valuable to the research community. However, there are several important items that must be addressed prior to publication. My major concerns are associated with a lack of resources that ensure reproducibility and others that facilitate the use of InterCellar for people without programming skills.

Major Comments:

Regarding reproducibility:

- Installing InterCellar in a clean R environment (version 4.1.0) using the command indicated in <https://github.com/martaint/InterCellar>

```
if (!requireNamespace("BiocManager", quietly = TRUE))
```

```
install.packages("BiocManager")
```

```
BiocManager::install("InterCellar")
```

After ~40 min of installation in a Macbook Pro, it output the following error:

```
ERROR: dependencies 'golem', 'ComplexHeatmap' are not available for package 'InterCellar'
```

Considering InterCellar is intended for scientist without programming expertise, I propose one of the following options to avoid this issue and other potential issues:

- o Creating an online website wherein scientists could easily upload their results and run the analyses without having to install InterCellar locally.

- o I understand that the previous point may not possible due to a lack of infrastructure for doing so or avoided because of data privacy. In that case, a web-platform could be replaced by a Docker container with InterCellar and all its dependencies pre-installed to facilitate its use and avoid dealing with potential issues associated to the installation.

- InterCellar is available in Bioconductor, but it would be useful having it also available in conda to ensure reproducible installations for scientists that use that platform.

Regarding the use of InterCellar:

- Although a tutorial with screenshots is provided

(http://bioconductor.org/packages/devel/bioc/vignettes/InterCellar/inst/doc/user_guide.html), an interactive tutorial/video would be more appropriate for guiding the use of InterCellar given its interactive functionalities (e.g., moving nodes in a cell-cell network, selecting items from tables, etc.).

- The authors provided the data and inputs needed to reproduce the examples in the manuscript (<https://github.com/martaint/InterCellar-reproducibility>); however, it is not clear how to use those files. A tutorial that shows how to use the files provided and how to generate the figures reported in the manuscript would be important to include. One option could be to replicate the tutorial in Bioconductor but showing how to use the input data for generating the results reported in the manuscript.

Regarding the examples and results reported:

- Figures 2-5 shows how the main results were obtained; however, one could ask how this is different from tools such as CellChat (<https://github.com/sqjin/CellChat>) and the visualization options it has. CellChat is a tool that requires more advanced programming skills, but it is not harder than using tools that are needed for preprocessing input data for InterCellar (e.g. Seurat and CellPhoneDB). With that said, I recommend adding to fig 2-5 some plots from CellChat or equivalent tools and demonstrate the visual improvement that InterCellar offers versus those visualizations (e.g. how the dotplots from InterCellar are better to interpret results, how the circos plot in Fig. 5b-e are different to the ones that CellChat can do, etc.). In other words, plot visualizations from one tool next to the other's and indicate when necessary the clear interpretation that InterCellar offers.

- Figure 5a is an interesting and novel visualization to represent groups of biological functions associated with cell-cell communication. However, it seems to be manually annotated from the table that InterCellar generated with the enriched functions. Since this figure was not automatically generated it may be misleading to include without providing a proper tutorial that shows how the annotations from the table were used for generating this figure (so users can have an idea on how they could do a similar figure).

- Some figures seem to require biological expertise to be generated, for example selecting which

ligand-receptor pairs to show in Fig. 3, or which functional term to consider in Fig 4. Given that, it would be great that InterCellar could include a data-driven approach to automatically select pertinent/important ligand-receptor interactions or functional annotations and generate an automatic visualization while offering the current way of manually selecting elements to show.

Minor comments:

- Figure 1 shows the workflow of InterCellar, which is useful to have an overall idea of what it is capable to do. I recommend visually improving that figure by clearly highlighting the steps (e.g. bold text or colors for numbers of each step)
- Cell types that are mentioned at the end of the first paragraph in "InterCellar highlights data-driven patterns of cell-cell communication in scRNA-seq data" can be omitted here and just included in Figure 2 caption.
- Interactions in Figure 2b are all types of flow like in Figure 2a? Also, it is not clear how it is different to consider just (directed-outgoing + directed-incoming) vs (directed-outgoing + directed-incoming + undirected), shouldn't undirected interactions include both directed-outgoing/incoming?
- For the discussion idea in page 20 "Since prediction methods often rely on different reference databases, and build their results on diverse statistical and mathematical assumptions¹⁰, evaluating advantages and disadvantages of each method is critical for choosing the method that best fits to the data of the user", I recommend adding this reference: <https://doi.org/10.1101/2021.05.21.445160>

Reviewer #3 (Remarks to the Author):

In "Interactive analysis and exploration of cell-cell communication in single-cell transcriptomics with InterCellar" the authors present an interactive platform for downstream analyses, visualisation and interpretation of cell-cell communication networks inferred from single-cell transcriptomics data. It is intended for biologists without much computational experience. As such, the platform is very well designed and user-friendly, it provides different options for users to explore and filter the data in an easy way and to apply different types of visualisation. It would be very helpful for biologists to explore and interpret the cell-cell communication results. However, aside from that, the platform does not offer anything novel in terms of methodology or computational approaches, many cell-cell communication tools offer similar analyses/visualisation options, with some computational skills required.

Specific comments:

- 1) It was not specified in the text how the authors annotated which proteins are receptors and ligands.
- 2) to make the usability broader more cell-cell communication methods should be added to the platform, in addition to the other two methods.
- 3) I liked the functional annotation of the ligand-receptor interactions, even though similar annotation has been done by other methods such as CellChat and SingleCellSignalR except there only Reactome

and KEGS have been used. However, from the analyses presented on the COVID-19 dataset, it wasn't clear to me what the relevance is of the fact that moderate cases had a higher total number of int-pairs annotated to "viral protection interaction with cytokine and cytokine receptor", or that critical cases had the highest number of unique int-pairs? Perhaps it would be more informative to also show the top biological processes that are enriched in severe vs moderate, and which ligand-receptor interactions are involved?

4) Some suggestions to introduce novelty would be perhaps new computational approaches to identify dysregulated interactions across conditions or to prioritise interesting interactions.

Point-by-point response to the reviewers' comments (1st round)

Interactive analysis and exploration of cell-cell communication in single-cell transcriptomics with InterCellar

Marta Interlandi, Kornelius Kerl and Martin Dugas

As a short note to the reviewers, we would like to mention that the new version (2.0.0) of InterCellar will be available with the new Bioconductor release from Wednesday, 27th of October.

Reviewers' comments:

Reviewer #1 (Remarks to the Author):

The authors present an interactive R/Shiny package, InterCellar, for downstream analysis of cell-cell communication inference (CCI) derived from scRNA-seq data. The goal of InterCellar is to analyze the data over three biological domains, referred to as “verses”: the (cell) cluster-verse, gene-verse, and the function-verse, which seeks to infer biological function via functional annotation of inferred interactions. In cluster-verse, InterCellar visualizes results with respect to the total number of interactions per cell type; in the gene-verse, InterCellar investigates which cluster-pairs communicate through which genes; and in the function-verse, it annotates cell-cell communication with biological functions implied by enriched pathways. The authors demonstrate its main features by applying it to two datasets of COVID-19 and melanoma from the literature. Using InterCellar, the authors present results that are consistent with previous findings; they also present some novel findings on cell communications in the two datasets. By comparison with similar packages, the authors claim that InterCellar has the following advantages: It provides a local and interactive analysis that focuses on enriched biological pathways without programming skills, and the workflow can be run over a relatively short time frame.

The manuscript is well written, particularly the Introduction. The figures are nicely done, and the motivation to streamline CCI analysis from single-cell RNA-seq data is worthwhile. Moreover, the functional analysis presented is a very interesting way to interpret CCI output. However, it is not fully clear what novel features InterCellar presents that are not available in other packages, such as CellChat. Moreover, the authors attribute many novel results not presented in previous studies to InterCellar's features, but one could arguably attribute the results to CellPhoneDB, which was applied to each dataset. Overall, there is a lack of

explanation of the underlying methods used during the downstream analysis and the arguments for why InterCellar should be adopted over other methods, particularly CellChat, could be fleshed out.

Therefore, I recommend acceptance only after major revisions.

Below are more comments:

1. It is great that the authors allow for flexibility of various CCI outputs as input for InterCellar. However, it is not entirely clear what type of data is needed. Is it a matrix, or an R dataframe? Clarifying this would strengthen the benefits of InterCellar more.

We thank the reviewer for pointing out this missing explanation. We have added a **Supplementary Note 1** clarifying which input data can be uploaded to InterCellar, depending on the different supported tools. In general, InterCellar's expected input is the output data directly generated by the supported tools, such as the folder (or file) where the CCI data has been saved. For example, InterCellar automatically parses the output folder generated by CellPhoneDB (containing 4 files: *deconvoluted.txt*, *means.txt*, *pvalues.txt*, and *significant_means.txt*) and extracts the relevant information necessary for the downstream analysis. Moreover, we have added a screenshot of InterCellar's custom data upload, that provides a table as an example of how to structure custom CCI data (**Supplementary Fig. 1**).

2. There is a lot of overlap between InterCellar and CellChat, in terms of the visualization capabilities. The way that the results are presented in the manuscript imply that InterCellar is a way to extract more meaning from results from packages like CellPhoneDB, as is done for CellChat. I think it would be worth fleshing out a comparison between the two methods to better highlight how InterCellar can be used for downstream analysis.

We agree that a comparison between InterCellar and CellChat is meaningful to clarify similarities and differences between the two tools. To this aim, we introduced, in this revised version, major changes in the structure of the Results section. Now, we focus first on illustrating InterCellar's features implemented in the three *universes* (*cluster-*, *gene-* and *function-verse*) and we take the opportunity to compare visualization outputs with the ones produced by CellChat (R package) in **Supplementary Note 3** (**Fig. 2** and **Supplementary Fig. 5**). Even though these two tools offer in general similar outputs and levels of customization, one of InterCellar's main advantages is the possibility to quickly and interactively explore the data with no coding needed. Moreover, the InterCellar web-based application allows the user to interact with dynamic outputs (e.g., networks/sunburst), while both the CellChat R package and Shiny app only offer static visualizations. These features become even more valuable when we consider the "target" end-user, for whom InterCellar was conceived (see the following comment). Finally, we present features that are novel to InterCellar (**Fig. 3** and **Supplementary Fig. 2**), specifically the definition of int-pair modules based on functional similarity (which we further compare to CellChat's functional similarity in **Supplementary Fig. 6** and **Supplementary Note 3**).

3. The authors rightfully stress the importance of reducing the "programming barrier" required to make better use of CCI packages. However, it is not clear how InterCellar is easier than other methods, as all cell-cell communication methods require some degree of programming

expertise. If researchers are able to run cell-cell communication analysis and analyze single-cell RNA-seq data, then it is not unreasonable to expect that they know how to visualize the communication output (with help from tutorials).

We are grateful to the reviewer for this comment, as it indicates that the “target” end-user of InterCellar was not clearly delineated in the manuscript. We implemented this interactive tool with the primary goal of fostering the collaboration between wet-lab scientists with great biological expertise (and less computational skills) and computer scientists, for whom the biological interpretation of complex cell-cell communication might not be trivial. Thus, the primary, target end-user of InterCellar might be a wet-lab scientist with no or little programming skills, who receives raw results of CCI packages from his/her bioinformatics collaborators. Although we acknowledge the fact that the one presented is a specific case, we hypothesize that this would not be a rare case in scientific research, where highly specialized professionals could greatly benefit from their complementary skills. Thus, InterCellar could be highly beneficial to clinicians and lab-scientists, by offering a user-friendly, interactive tool that would help streamline the downstream analysis of cell-cell communication. To better explain this point in the manuscript, we added a sentence in the Introduction (lines 85-88).

4. The filtering/visualisation steps of InterCellar are very nice but there's not much mention of them in the main results text. It would be interesting to see how the filtering used by InterCellar affects downstream analysis.

We thank the reviewer for the interest in the filtering options of InterCellar. Although we acknowledge the fact that applying any filtering step on CCI data will affect the downstream results, we believe that a thorough evaluation of different filtering schemes may be beyond the scope of this manuscript. Instead, we have added a detailed explanation of filtering options available, depending on the input data, in InterCellar's gene-verse (**Supplementary Note 2**). Moreover, we have better specified in the text the idea behind these filters (lines 148-151): InterCellar provides dynamic filters that can be manually and interactively set by the users, upon specific needs (e.g., removing a subset of cell types or int-pairs with score < threshold).

5. It is unclear whether the figures shown in the manuscript are produced near-automatically from the R/Shiny workflow of InterCellar, or if these are polished figures based on raw output from InterCellar. If the former is true, this should be highlighted more to strengthen the novelty of InterCellar. If the latter is true, then this is somewhat contradictory to the aims of InterCellar to lower the programming expertise barrier.

We are sorry for not pointing this out more clearly in the manuscript. The figures shown in the **original** manuscript (**Fig. 2-5** and **Supplementary Fig. 1-3**) are generated automatically by InterCellar. Only minor polishing has been added to the aforementioned figures (with the exception of **Fig. 5a**, see the response to comment 5., Reviewer #2), using Adobe Illustrator (e.g., text enlargement, box addition in sunburst plots). The R/Shiny app offers multiple options for downloading visualizations (e.g., html, tiff, or pdf formats), with the aim of facilitating the collection of “publication-ready” results. We added a sentence in each figure legend to clearly mention this.

6. Counting only number of interactions/interaction pairs assumes that each pair has equal effect on communication. Have the authors considered weighting the interactions by interaction score, i.e. instead of number of interactions, you count total interaction score, and instead of Figure 4 counting the fraction of numbers, you could the fraction of total weight?

We implemented a new feature in InterCellar following the reviewer's suggestion. In particular, we added the option to see either (i) number of interactions or (ii) weighted number of interactions (weighted by the interaction score) in multiple visualization options: *cluster-verse* network and barplot, *gene-verse* network, and *function-verse* sunburst plot. We also updated **Fig. 4** (now **Fig. 6**) sunburst plots by showing the weighted number of interactions.

7. The function-verse is perhaps the most novel feature of InterCellar and is a very nice way to interpret CCI output. I think the descriptions underlying the function-verse-specific methods could be expanded more to highlight its novelty and usefulness.

To address this point, we added a detailed explanation of InterCellar's sunburst plot in the Results chapter "InterCellar allows data exploration through a user-friendly interface, customization options, and interactive visualizations".

8. The authors use the function-verse outputs to classify interactions based on functional similarity. The term "functional similarity" is also used in CellChat for downstream analysis, albeit using a different methodology. Again, it would be useful to highlight the differences between InterCellar and CellChat for functional analysis.

Indeed, both CellChat and InterCellar use "functional similarity" in the downstream analysis. However, the results are very different since the two methods differ in what is defined to be functionally similar. To explain this point we have added **Supplementary Fig. 2** and **6**, which depict InterCellar's methods to compute functional similarity as well as the output of CellChat's functional similarity analysis. Moreover, we explain in detail the differences between the two methods in **Supplementary Note 3**. Briefly, InterCellar seeks to calculate a similarity between int-pairs, based on the functional terms that have been annotated to them. The result is therefore groups of int-pairs that are similar in their biological functions ("int-pair modules"). On the contrary, CellChat considers the similarity between signaling pathways with the aim of identifying groups of pathways sharing similar "sender" and "receiver" clusters. Thus, the two approaches might be considered complementary, offering two points of view on the concept of functional similarity.

9. As the authors run CellPhoneDB for each of the datasets considered, it is hard to attribute any of the new biological results to InterCellar specifically, as there is no benchmark of comparison. Moreover, none of the original papers had run CellPhoneDB, so it is not entirely clear whether InterCellar would have found new results that would not have been found if users had just applied CellPhoneDB on its own. Is it possible to analyse a dataset where the CCI has already been run, where the authors do not need to run CellPhoneDB?

In this revised version of the manuscript, we run CellChat on the melanoma dataset, to fairly compare the two methods (**Fig. 2** and **Supplementary Fig. 5**) and to demonstrate the general usability of InterCellar, whose workflow does not depend on the input method chosen. However, as Chua et al. used CellPhoneDB in their original manuscript on COVID-19, we did

not change that and simply added a sentence in the manuscript (line 307). With regards to the attribution of new biological results to InterCellar, we have shown that the advantage of our tool is the possibility to conduct an unbiased analysis. On the contrary, as many CCI-inference tools offer very limited downstream analysis options, the biological interpretation is prone to be biased by the analyst's previous knowledge. In the specific case of the COVID-19 dataset, the higher signalling activity of mast cells in critical cases (compared to moderate) could be detected thanks to the unbiased and systematic consideration of all cell-clusters in the datasets.

10. The authors state that InterCellar accepts input from SingleCellSignalR and custom CCI input. However, the results presented only consider CellPhoneDB-derived output. How do the results change if SingleCellSignalR or other CCI output is used instead?

We appreciate the reviewer's interest in InterCellar's input features. The main purpose of supporting multiple published CCI inference methods as input to InterCellar is to provide a flexible analysis platform that can be widely used, independently of one's favorite inference method. For this reason, we extended the list of supported tools, including CellChat and ICELLNET. However, comparing InterCellar results on different input methods would inevitably translate to a comparison of the input methods themselves, thus going beyond the scope of this paper. In this regard, following the suggestion of Reviewer #2 (comment 11), we have added in the Discussion a reference to a benchmarking paper (<https://doi.org/10.1101/2021.05.21.445160>) that systematically compares multiple CCI inference tools. Finally, we want to mention that InterCellar's workflow and features are conserved across multiple input methods, thus providing a general but robust analysis platform.

11. CellPhoneDB handles multiple ligand/receptor sub-units but it looks like InterCellar does not. How does InterCellar reconcile this?

We are thankful to the reviewer for this comment, which prompted us to describe in greater detail how the data from different input methods is handled (**Supplementary Note 1**) as well as add a sentence in the Methods' chapter "Functional annotation of interaction pairs" to explain how the functional annotation is performed in the case of multiple ligand/receptor sub-units. In summary, InterCellar retains the information related to multiple sub-units and complexes, when these are available from the input data of the selected tool (e.g., CellPhoneDB and CellChat)

12. I see that on BioConductor, InterCellar requires R 4.1, but in the manuscript, the authors claim that there are installation instructions for R 4.0.X at the GitHub repository. However, I could not find these instructions.

We are sorry for this, we had to remove the option to install InterCellar with R4.0 after submission of the manuscript, as R/Bioconductor's new release became available and we wanted to avoid possible package versions issues. We have deleted this sentence in the manuscript.

Reviewer #2 (Remarks to the Author):

Existing tools for inferring cell-cell communication often require computational expertise to interpret biological signals, which limits scientists without programming skills to easily analyze, explore and interpret predicted intercellular communication. In this work, Interlandi, Kerl and Dugas present InterCellar, an interactive platform intended to fulfill this gap and facilitate the biological interpretation through customizable analyses and visualization options to easily identify important ligand-receptor interactions and meaningful associated biological pathways. The authors also provided COVID-19 and cancer examples to demonstrate what InterCellar offers for achieving this purpose. This work claims to empower lab-scientist without programming skills to analyze and explore results from inferred cell-cell communication. Meeting this intention is an important contribution to the cell-cell communication field, which would clearly help massify these analyses and have a better understanding of associated biological processes, this this work will be valuable to the research community. However, there are several important items that must be addressed prior to publication. My major concerns are associated with a lack of resources that ensure reproducibility and others that facilitate the use of InterCellar for people without programming skills.

Major Comments:

Regarding reproducibility:

1. Installing InterCellar in a clean R environment (version 4.1.0) using the command indicated in <https://github.com/martaint/InterCellar>

```
if (!requireNamespace("BiocManager", quietly = TRUE))
  install.packages("BiocManager")
```

```
BiocManager::install("InterCellar")
```

After ~40 min of installation in a Macbook Pro, it output the following error:

```
ERROR: dependencies 'golem', 'ComplexHeatmap' are not available for package 'InterCellar'
```

Considering InterCellar is intended for scientist without programming expertise, I propose one of the following options to avoid this issue and other potential issues:

- o Creating an online website wherein scientists could easily upload their results and run the analyses without having to install InterCellar locally.

- o I understand that the previous point may not possible due to a lack of infrastructure for doing so or avoided because of data privacy. In that case, a web-platform could be replaced by a Docker container with InterCellar and all its dependencies pre-installed to facilitate its use and avoid dealing with potential issues associated to the installation.

It's unfortunate to learn that installation via Bioconductor was not successful. We believed that this solution could have been the safest one as Bioconductor usually takes care of

dependencies and versioning. We followed the reviewer's suggestion of providing a docker environment alongside the Bioconductor package. Installation instructions can be found in InterCellar's GitHub repository. Specifically, the docker container provided is linked to the release version of the Bioconductor package, to have consistent packages in different platforms (see also the following comment). We decided not to provide an online website as this would create issues for data privacy.

2. InterCellar is available in Bioconductor, but it would be useful having it also available in conda to ensure reproducible installations for scientists that use that platform.

We thank the reviewer for this suggestion. We have now added a link in the manuscript, pointing to the bioconda recipe which is automatically generated for each Bioconductor package (<http://bioconda.github.io/recipes/bioconductor-intercellar/README.html>).

Regarding the use of InterCellar:

3. Although a tutorial with screenshots is provided (http://bioconductor.org/packages/devel/bioc/vignettes/InterCellar/inst/doc/user_guide.html), an interactive tutorial/video would be more appropriate for guiding the use of InterCellar given its interactive functionalities (e.g., moving nodes in a cell-cell network, selecting items from tables, etc.).

We thank the reviewer for giving us this idea. We created a video-tutorial of InterCellar which can be found at <https://uni-muenster.sciebo.de/s/23xifn3re3QbSRC>.

4. The authors provided the data and inputs needed to reproduce the examples in the manuscript (<https://github.com/martaint/InterCellar-reproducibility>); however, it is not clear how to use those files. A tutorial that shows how to use the files provided and how to generate the figures reported in the manuscript would be important to include. One option could be to replicate the tutorial in Bioconductor but showing how to use the input data for generating the results reported in the manuscript.

We agree that such a tutorial would be important and highly beneficial to the reproducibility of the results shown in this paper. We added this and further documentation in the InterCellar-reproducibility GitHub repository (<https://github.com/martaint/InterCellar-reproducibility>), specifically in the intercellar-walkthrough folder.

Regarding the examples and results reported:

5. Figures 2-5 shows how the main results were obtained; however, one could ask how this is different from tools such as CellChat (<https://github.com/sqjin/CellChat>) and the visualization options it has. CellChat is a tool that requires more advanced programming skills, but it is not harder than using tools that are needed for preprocessing input data for InterCellar (e.g. Seurat and CellPhoneDB). With that said, I recommend adding to fig 2-5 some plots from CellChat or equivalent tools and demonstrate the visual improvement that InterCellar offers versus those visualizations (e.g. how the dotplots from InterCellar are better to interpret results, how the circos plot in Fig. 5b-e are different to the ones that CellChat can do, etc.). In other words, plot visualizations from one tool next to the other's and indicate when necessary the clear interpretation that InterCellar offers.

With regards to the first part of the comment, we refer the reviewer to the response to comment 3., reviewer #1. There we delineated in detail InterCellar's primary end-user.

Regarding the comparison with CellChat, we refer the reviewer to the answers to comments 2. and 8. of reviewer #1.

6. Figure 5a is an interesting and novel visualization to represent groups of biological functions associated with cell-cell communication. However, it seems to be manually annotated from the table that InterCellar generated with the enriched functions. Since this figure was not automatically generated it may be misleading to include without providing a proper tutorial that shows how the annotations from the table were used for generating this figure (so users can have an idea on how they could do a similar figure).

We thank the reviewer for this comment, which allowed us to better explain how the manual annotation of the interaction-pairs UMAP can be achieved. Specifically, this can be found in the updated Bioconductor user-guide, and briefly mentioned in **Fig. 3** legend.

7. Some figures seem to require biological expertise to be generated, for example selecting which ligand-receptor pairs to show in Fig. 3, or which functional term to consider in Fig 4. Given that, it would be great that InterCellar could include a data-driven approach to automatically select pertinent/important ligand-receptor interactions or functional annotations and generate an automatic visualization while offering the current way of manually selecting elements to show.

We are grateful for this comment, which highlighted that the data-driven part of InterCellar analysis was not presented and stressed clearly enough in the original manuscript. We took great care in this revised version to amend this point. First of all, we restructured the Results section by 1) demonstrating InterCellar's data exploration features, available in the three *universes* and 2) illustrating the data-driven analysis part, which is now composed of two separate sections, "int-pair modules" and comparison of "multiple conditions". Both sections offer data-driven results that are designed to facilitate biological interpretation and streamline the analysis of CCI data.

Minor comments:

8. Figure 1 shows the workflow of InterCellar, which is useful to have an overall idea of what it is capable to do. I recommend visually improving that figure by clearly highlighting the steps (e.g. bold text or colors for numbers of each step)

We are grateful to read the reviewer's appreciation for Figure 1. We have now changed the color and increased the size of the 3 main steps of the workflow.

9. Cell types that are mentioned at the end of the first paragraph in "InterCellar highlights data-driven patterns of cell-cell communication in scRNA-seq data" can be omitted here and just included in Figure 2 caption.

Good point, we have adjusted this in the manuscript.

10. Interactions in Figure 2b are all types of flow like in Figure 2a? Also, it is not clear how it is different to consider just (directed-outgoing + directed-incoming) vs (directed-outgoing + directed-incoming + undirected), shouldn't undirected interactions include both directed-outgoing/incoming?

Yes, interactions considered for the original **Fig. 2b** (now **Fig. 4b**) are all types of flow. Undirected interactions, specifically, account for R-R and L-L pairs, which can be found in certain input methods. These are an important addition to directed interactions (L-R pairs) for data generated by CellPhoneDB.

11. For the discussion idea in page 20 "Since prediction methods often rely on different reference databases, and build their results on diverse statistical and mathematical assumptions¹⁰, evaluating advantages and disadvantages of each method is critical for choosing the method that best fits to the data of the user", I recommend adding this reference: <https://doi.org/10.1101/2021.05.21.445160>

We thank the reviewer for this addition. Indeed, we found this paper of great interest and added this reference in the Introduction and Discussion.

Reviewer #3 (Remarks to the Author):

In "Interactive analysis and exploration of cell-cell communication in single-cell transcriptomics with InterCellar" the authors present an interactive platform for downstream analyses, visualisation and interpretation of cell-cell communication networks inferred from single-cell transcriptomics data. It is intended for biologists without much computational experience. As such, the platform is very well designed and user-friendly, it provides different options for users to explore and filter the data in an easy way and to apply different types of visualisation. It would be very helpful for biologists to explore and interpret the cell-cell communication results. However, aside from that, the platform does not offer anything novel in terms of methodology or computational approaches, many cell-cell communication tools offer similar analyses/visualisation options, with some computational skills required.

Specific comments:

1. It was not specified in the text how the authors annotated which proteins are receptors and ligands.

We thank the reviewer for pointing out this missing explanation. We have now added a **Supplementary Note 1** describing how the input data from different tools is handled by InterCellar and in particular, how the annotation to ligand and receptor is carried out.

2. to make the usability broader more cell-cell communication methods should be added to the platform, in addition to the other two methods.

We completely agree and we added two more methods (CellChat and ICELLNET) to the list of supported tools.

3. I liked the functional annotation of the ligand-receptor interactions, even though similar annotation has been done by other methods such as CellChat and SingleCellSignalR except there only Reactome and KEGS have been used. However, from the analyses presented on the COVID-19 dataset, it wasn't clear to me what the relevance is of the fact that moderate cases had a higher total number of int-pairs annotated to "viral protection interaction with cytokine and cytokine receptor", or that critical cases had the highest number of unique int-pairs? Perhaps it would be more informative to also show the top biological processes that are enriched in severe vs moderate, and which ligand-receptor interactions are involved?

We are grateful to the reviewer for this comment, which prompted us to extend InterCellar's functionalities in this revised version. In particular, we have now implemented functionalities to upload multiple CCI data and (i) conduct a parallel analysis for data exploration (in *universes*) and definition of "int-pair modules" and (ii) compare multiple conditions to highlight differences in terms of cell-cell communication and functional pathways. This is reflected in InterCellar's third step of the workflow which has been renamed "Data-driven analysis" and holds the two sections for int-pair module analysis and multiple conditions comparison. With regards to the specific comment, we have now updated the original **Fig. 4** (now **Fig. 6**) by showing the new results that can be obtained in the function-based "multiple conditions" section (chapter "InterCellar uncovers condition-specific interactions and their related biological pathways"). Briefly, InterCellar compares here two conditions (in our case, COVID-19 critical vs moderate), determines which int-pairs are distinctively found in each condition, and finally performs a one-sided permutation test to determine which functional terms are significantly annotated in one condition *versus* the other. Significant terms are then displayed in a table and can be visualized as a sunburst plot, delivering an information-rich output that highlights dysregulated pathways across conditions.

4. Some suggestions to introduce novelty would be perhaps new computational approaches to identify dysregulated interactions across conditions or to prioritise interesting interactions.

Once again, we thank the reviewer and refer to the previous answer.

REVIEWERS' COMMENTS:

Reviewer #1 (Remarks to the Author):

We thank the authors for their efforts to address the previous comments. InterCellar presents additional features that complement CellChat and other cell-cell communication inference packages. It considers another approach when calculating the similarity of int-pairs, which refers to whether they perform similar functions biologically, as defined by their association with functional terms pulled from curated ontology databases. This novelty is nice and could be complementary to the existing tools. However, compared with existing methods, this appears to be the only piece of novelty in terms of computational approaches in InterCellar. Another concern is that this paper does not provide any new biological findings, but rather proposes a new interactive tool. As such, this journal may not be the best fit for this paper.

Below are a few additional detailed comments.

1. The screenshot of results from Shiny app in the tutorial is in a very low resolution, making it very difficult to read.
2. Although the authors claimed that InterCellar now supports the output of computational tools including CellChat and ICELLNET, the selected tool in the Shiny app still does not support the input of these two added tools.
3. In the Abstract, the authors claimed that "InterCellar implements data-driven analyses including the possibility to compare cell-cell communication from multiple conditions". However, the comparison analysis in InterCellar is only about identification of functional terms.

Reviewer #2 (Remarks to the Author):

The authors addressed all of our comments. We look forward to seeing this work in print

Reviewer #3 (Remarks to the Author):

The authors addressed all of my comments.

Point-by-point response to the reviewers' comments (2nd round)

InterCellar enables interactive analysis and exploration of cell-cell communication in single-cell transcriptomic data

Marta Interlandi, Kornelius Kerl and Martin Dugas

REVIEWERS' COMMENTS:

Reviewer #1 (Remarks to the Author):

We thank the authors for their efforts to address the previous comments. InterCellar presents additional features that complement CellChat and other cell-cell communication inference packages. It considers another approach when calculating the similarity of int-pairs, which refers to whether they perform similar functions biologically, as defined by their association with functional terms pulled from curated ontology databases. This novelty is nice and could be complementary to the existing tools. However, compared with existing methods, this appears to be the only piece of novelty in terms of computational approaches in InterCellar. Another concern is that this paper does not provide any new biological findings, but rather proposes a new interactive tool. As such, this journal may not be the best fit for this paper.

Below are a few additional detailed comments.

1. The screenshot of results from Shiny app in the tutorial is in a very low resolution, making it very difficult to read.

We apologize for this inconvenience, unfortunately, we are limited by the Bioconductor requirements on the file size of the InterCellar user guide. However, we have generated higher resolution screenshots for the InterCellar-walkthrough on COVID-19 and melanoma, which can be found in the GitHub repository (<https://github.com/martaint/InterCellar-reproducibility/>).

2. Although the authors claimed that InterCellar now supports the output of computational tools including CellChat and ICELLNET, the selected tool in the Shiny app still does not support the input of these two added tools.

We thank the reviewer for this comment and we would appreciate if the reviewer would take a second look at this matter. Below, we provide a screenshot of the expanded drop-down menu of the "Selected tool" section of the Shiny app. An example showing the selection of CellChat can be found in **Supplementary Figure 1**.

Analysis setup

Please select an existing local folder where InterCellar will save all results of your analysis.

From supported tools From custom analysis

CCI data #1

CCI data ID

Output folder tag

Select tool:

CellPhoneDB v2
CellChat
ICELNET
SingleCellSignalR

CCI data #2

3. In the Abstract, the authors claimed that” InterCellar implements data-driven analyses including the possibility to compare cell-cell communication from multiple conditions”. However, the comparison analysis in InterCellar is only about identification of functional terms.

Once again, we are grateful for this comment and we kindly refer the reviewer to **Figure 1** for an overview of the different features of the “multiple conditions” module of InterCellar. These can be then appreciated in detail in **Figures 4, 5, and 6**. In brief, the “multiple conditions” module provides comparative analyses in terms of (1) total and relative number of interactions, through bar plots and radar plots; (2) condition-specific “int-pair/cluster-pair couplets”, visualized in dot plots and summarized by pie charts; and (3) condition-specific interactions and their enriched significant functional terms, summarized in sunburst plots.

Reviewer #2 (Remarks to the Author):

The authors addressed all of our comments. We look forward to seeing this work in print

Reviewer #3 (Remarks to the Author):

The authors addressed all of my comments.